# Reconstruction of zinc-metal battery solvation structures operating from −50 ~ +100 °C

**Lingbo Yao**[1,2], **Jiahe Liu**[1,2], **Feifan Zhang**[1,2], **Bo Wen**[1,2], **Xiaowei Chi** ◎[1] ✉ & **Yu Liu** ◎[1] ✉

Serious solvation effect of zinc ions has been considered as the cause of the severe side reactions (hydrogen evolution, passivation, dendrites, and etc.) of aqueous zinc metal batteries. Even though the regulation of cationic solvation structure has been widely studied, effects of the anionic solvation structures on the zinc metal were rarely examined. Herein, co-reconstruction of anionic and cationic solvation structures was realized through constructing a new multi-component electrolyte ($Zn(BF_4)_2$-glycerol-boric acid-chitosan-poly-acrylamide, simplified as ZGBCP), which incorporates double crosslinking network via the esterification, protonation and polymerization reactions, thereby combining multiple advantages of 'liquid-like' high conductivity, 'gel-like' robust interface, and 'solid-like' high $Zn^{2+}$ transfer number. Based on the ZGBCP electrolyte, the Zn anodes achieve record-low polarization and stable cycling. Furthermore, the ZGBCP electrolyte renders the AZMBs ultrawide working temperature (−50 °C ~ +100 °C) and ultralong cycle life (30000 cycles), which further validates the feasibility of the dual solvation structure strategy and provides a innovative perspective for the development of high-performance AZMBs.

Aqueous zinc metal batteries (AZMBs) with considerable advantages such as high theoretical energy density, environmental friendliness, intrinsic flame retardancy, and low cost, are expected to play a significant role in future advanced energy-storage applications[1]. However, traditional AZMBs fail to operate reliably in practical conditions due to their poor cycling stability and narrow operating temperature windows, which result from the severe side reactions of the zinc metal anode and the high freezing point of the traditional sulfate electrolyte[2]. Fundamentally, the side reactions are caused by the strong solvation effect of water molecules on the $Zn^{2+}$ and the corrosion induced by sulfate anions[3]. Therefore, reconstructing the solvation structures of zinc cations and especially reconstructing/replacing sulfate anions simultaneously are the ideal solutions.

First, to regulate the solvation structures of zinc cations, different organic solvents (e.g. ethylene glycol, sulfolane, and trimethyl phosphate) with stronger coordinating capability were introduced into the electrolyte[4-6]. Organic molecules can replace partial water molecules of the $Zn^{2+}$ solvation shell, which can suppress the water-induced side reactions and enhance the stability of stripping/plating reactions[4-6]. In addition, highly concentrated electrolytes were also developed to decrease the activity of water molecules and improve the stability of Zn anode[7]. However, the reaction kinetics and especially the cathode's performance were sacrificed due to the considerably increased viscosity and decreased ionic conductivity. Second, compared to traditional sulfate salts, these special salts (e.g. $CF_3SO_3^-$, $(CF_3SO_2)_2N^-$, etc.) are more friendly to the Zn anode due to certain solid electrolyte interface (SEI) formation. However, the interactions between the anions and cations inevitably lead to the formation of ionic aggregates (AGG) at high concentrations, which may further result in slow diffusion and desolvation kinetics under low

[1]Shanghai Institute of Ceramics, Chinese Academy of Sciences, 200050 Shanghai, China. [2]University of Chinese Academy of Sciences, 100049 Beijing, China. ✉e-mail: xwchi@mail.sic.ac.cn; yuliu@mail.sic.ac.cn

temperatures as shown in Scheme 1a[8]. Reflecting on the advancements in solvation structure design for AZMBs, there exists a notable deficiency in effective strategies for both anodes and cathodes. Unfortunately, little research has been reported to attempt to elaborate on the anionic solvation structures. Significant challenges persist regarding the impact of free anionic solvation structures dominated by weak intermolecular interactions on the properties of aqueous electrolytes and the stability of the interface[9–11]. Therefore, it's crucial to develop a new electrolyte that achieves co-reconstruction of cationic and anionic solvation structures and benefits the stability of both the Zn anode and cathode in a wide working temperature range.

Herein, based on the unique boron-polyol chemistry[12], a new strategy of co-reconstruction of anionic and cationic solvation structures (CRACSS) is proposed, leading to the design of a new quinary electrolyte ZGBCP, which endows the AZMBs both fast reaction kinetics and especially the so-far reported widest temperature range (−50 °C ~ +100 °C). The ZGBCP electrolyte incorporates a dual cross-linking network via three types of reactions: esterification, protonation, and polymerization reactions, which integrates the high conductivity (28.7 mS cm⁻¹ under 20 °C), intimate interface (778 mN m⁻¹ adhesion work) and high Zn²⁺ transfer number ($t_{Zn2+}$ = 0.82). Based on the combined advantages, the Zn||Zn symmetric cells exhibit an ultralow polarization voltage of 268 mV at an ultrahigh current density of 30 mA cm⁻² and stable cycling exceeding 3000 h at 1 mA cm⁻². Moreover, through re-constructing the cationic and anionic solvation structures, the ZGBCP electrolyte-based AZMBs ensure ultralong cycle life (30,000 cycles). These results represent notable advancements in the stability of AZMBs under high temperatures, which supports the potential effectiveness of the CRACSS strategy and provides a new direction for enhancing AZMBs' performance.

## Results

### Molecular design, fabrication, and properties of the new quinary ZGBCP electrolyte

As illustrated in Fig. 1a, water molecules show a strong solvation effect on the Zn²⁺, which has been demonstrated to be the main cause of hydrogen evolution reaction (HER) and dendrite formation in traditional ZnSO₄ aqueous electrolyte[13]. Same phenomenon is observed in the Zn(BF₄)₂-based aqueous electrolyte (AE)[14]. In particular, except for the solvation effect of Zn²⁺ cations, the water molecules can attack BF₄⁻ anions and lead to hydrolysis of the anions based on the reaction (1), thus exacerbating corrosion and HER side reactions at the Zn anode/electrolyte interface[15].

$$BF_4^- + H_2O \leftrightarrow BF_3(OH)^- + F^- + H^+ \tag{1}$$

Even though the solvation effect from water can be alleviated based on the organic solvent replacement according to the previously reported strategies, new issues of sluggish kinetics for Zn²⁺ at low temperatures and corrosion/decomposition for the BF₄⁻ anions at high temperatures have arisen[16]. To address the challenges as shown in Fig. 1a, a new quinary electrolyte system, ZGBCP hydrogel, was developed to realize the regulation of both Zn²⁺ cations and BF₄⁻ anions. First, glycerol, chitosan (CS), and polyacrylamide (PAM) with rich O and N atoms can form multiple strong hydrogen bonding with water molecules and influence the solvation structure of Zn²⁺ cations. Second, boric acid is capable of capturing HF and F⁻, thus suppressing the corrosion induced by BF₄⁻ anions based on the reaction such as (2)[17–19]:

$$BF(OH)_3^- \leftrightarrow B(OH)_3 + F^- \tag{2}$$

Moreover, three types of reactions: reversible esterification reaction between glycerol and boric acid (Supplementary Fig. 1)[20, 21],

protonation of chitosan and polymerization of acrylamide are involved in the synthetic process of the ZGBCP electrolyte. The delicate molecular design renders the electrolyte multiple unique properties: 'liquid-like' high conductivity, 'gel-like' robust interfacial adhesion, 'solid-like' high Zn²⁺ transfer number and robust mechanical strength (Fig. 1b).

First, as seen in Fig. 2a, the ZGBCP electrolyte exhibits a conductivity of 28.7 mS cm⁻¹ at room temperature, which is comparable to the AE (34.11 mS cm⁻¹), and it maintains high conductivity over a wide temperature range from −50 to +80 °C. The unique hydrogen bonding network also endows the electrolyte superior water retention of 97.5% after exposing to the ambient air for nearly 50 days (Fig. 2b) as well as exceptional thermal stability towards extreme temperature and flaming conditions (Supplementary Fig. 3). Additionally, the designed dual cross-linking network with multiple hydrogen bonding network exhibit significant energy dissipation behavior through deformation flow, since the boric acid-glycerol reaction can facilitate the polymer mobility effectively as illustrated in Fig. 2c[22, 23]. This feature is advantageous for facilitating rapid ion transport and mitigating the volumetric expansion of the Zn anodes during the high-rate charge and discharge process[24, 25].

Moreover, the ZGBCP electrolyte achieves robust interfacial adhesion effortlessly to many common materials such as zinc, stainless steel, carbon cloth, carbon paper, and polyaniline (PANI) with a large area of 5 × 5 cm² but a tiny contact area of 0.5 cm² (Fig. 2d and Supplementary Fig. 4). Remarkably, even at −40 and 80 °C, the adhesion shows no change (Fig. 2d). The adhesion force of the ZGBCP to the Zn electrode reaches as high as 8.9 kPa, which is higher than other two electrolytes (Supplementary Fig. 5). Moreover, the optical microscope images reveal that ZGBCP electrolyte shows much better adhesion interface than the ZGP and ZCP electrolytes (Fig. 2e and Supplementary Fig. 6), which is consistent with the highest adhesion work (778 mN m⁻¹) characterized by the atomic force microscopy testing (Fig. 2f). Fundamentally, this can be attributed to the multiple electrostatic and weak intermolecular interactions induced by the esterification and protonation process in the ZGBCP electrolyte (Fig. 2g)[26–29]. Besides, the faster motion of the chain segment tuned by the esterification also helps the adhesive groups (−OH, −NH₂, and −CONH₂) diffuse dynamically to form a consistently robust adhesion. In addition, the polymerization of AM can enhance the crosslinking density, ensuring the reinforced stability between the electrolyte and the electrode. In summary, all three reactions are essential for the robust interface of the ZGBCP electrolyte with electrodes.

### Theoretical calculation and characterizations of the ZGBCP electrolyte and understanding of CRACSS strategy in solvation regulation

To gain insights into the CRACSS strategy and correlate the structure with the property of the ZGBCP electrolyte, theoretical calculation and experimental characterizations of the molecular and solvation structure of the cations and anions were performed. Firstly, detailed density functional theory (DFT), together with molecular dynamics (MD) simulations were conducted to investigate the impact of the CRACSS strategy on the solvation structure. For the cationic solvation structure, the PAM (−1.24 eV) and glycerol (−1.09 eV) show higher affinity to Zn²⁺ than H₂O (−0.64 eV) and even near to the binding energy of 2H₂O−Zn²⁺, which enables the reconstruction of the cationic solvation structure through replacing H₂O to suppress the HER side reactions (Fig. 3a and Supplementary Fig. 7b). According to MD simulations, compared to the solvation structure Zn(BF₄⁻)₀.₈(H₂O)₅.₂ found in the AE, a new cationic coordination configuration (Zn²⁺(BF₄⁻)₁.₄(H₂O)₃.₆(C₃H₈O₃)₀.₆PAM₀.₄, simplified as [Zn²⁺(BF₄⁻)(H₂O)₄(C₃H₈O₃)]⁺-PAM is formed in the ZGBCP electrolyte (Fig. 3b and Supplementary Fig. 8). Besides, the reductive tendency of H₂O can be effectively suppressed, due to the much more balanced electrostatic potential (ESP) distribution of the distal coordinated H₂O in the new configuration (Fig. 3c). Furthermore, the

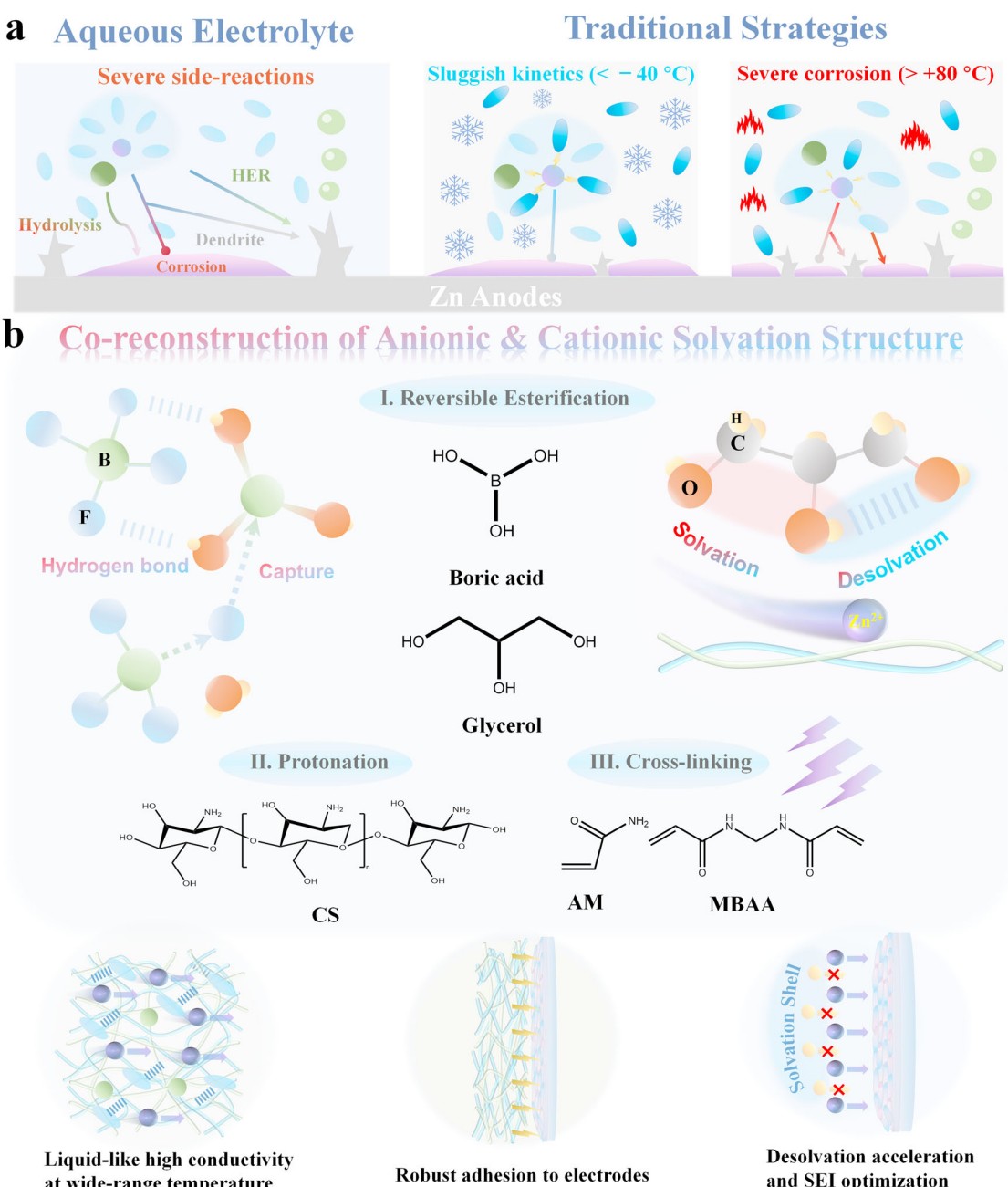

**Fig. 1 | Challenges of traditional aqueous electrolytes and unique characteristics of the ZGBCP electrolyte. a** The interfacial issues of the aqueous electrolyte and traditional modification strategies. **b** Molecular design, fabrication, and unique properties of the ZGBCP electrolyte.

DFT calculations indicate that the electrostatic interaction energies (denoted as the $H^+$ dissociation energy barrier) for H in the $H_2O$ of the solvation structure have been increased by approximately 2 eV compared to the $[Zn^{2+}(BF_4^-)(H_2O)_5]^+$ referring to Supplementary Fig. 8c, which also proves the suppressed HER tendency tuned by the reconstruction of the cationic solvation structure. In addition, the Supplementary Fig. 10 reveals the lowest solvation energy of $[Zn^{2+}(BF_4^-)(H_2O)_4(C_3H_8O_3)]^+$-PAM (−9.11 eV) compared to other potential solvation structures, confirming the validity of MD results. Another two potential solvation structures without the polymer involved: $[Zn^{2+}(H_2O)_5(C_3H_8O_3)]^{2+}$ (solvent-separated ion pairs, SSIP) and $[Zn^{2+}(BF_4^-)(H_2O)_4(C_3H_8O_3)]^+$ (CIP) also show lower HER tendency in comparison with $[Zn^{2+}(BF_4^-)(H_2O)_5]^+$ and $[Zn^{2+}(H_2O)_6]^{2+}$ in the AE as illustrated in Supplementary Figs. 11 and 12. Although it is a small amount for CS, a new AGG solvation structure

$[Zn^{2+}(BF_4^-)_2(H_2O)_2(C_3H_8O_3)]$-CS was found near the CS chain segment due to its stronger affinity to both cations and anions, which is conducive to the desolvation process with highest $H^+$ dissociation energy barrier (Supplementary Fig. 13). What's more, as illustrated in Supplementary Fig. 14, the results of MD simulations based on the PAM and CS pentamer models are consistent with those results based on the trimer models, further verifying the validity of MD simulations.

The solvation environment of anions also has a significant impact on the properties of electrolytes. Fundamentally, the anionic solvation structure is primarily governed by weak interactions such as hydrogen bonding. As seen in Fig. 3a and Supplementary Fig. 15, the glycerol ($C_3H_8O_3$) and boric acid ($H_3BO_3$) molecules show higher binding energy (−0.30 eV for $C_3H_8O_3$ and −0.32 eV for $H_3BO_3$) and higher bond order of H···F (0.171 for $C_3H_8O_3$ and 0.178 for $H_3BO_3$)

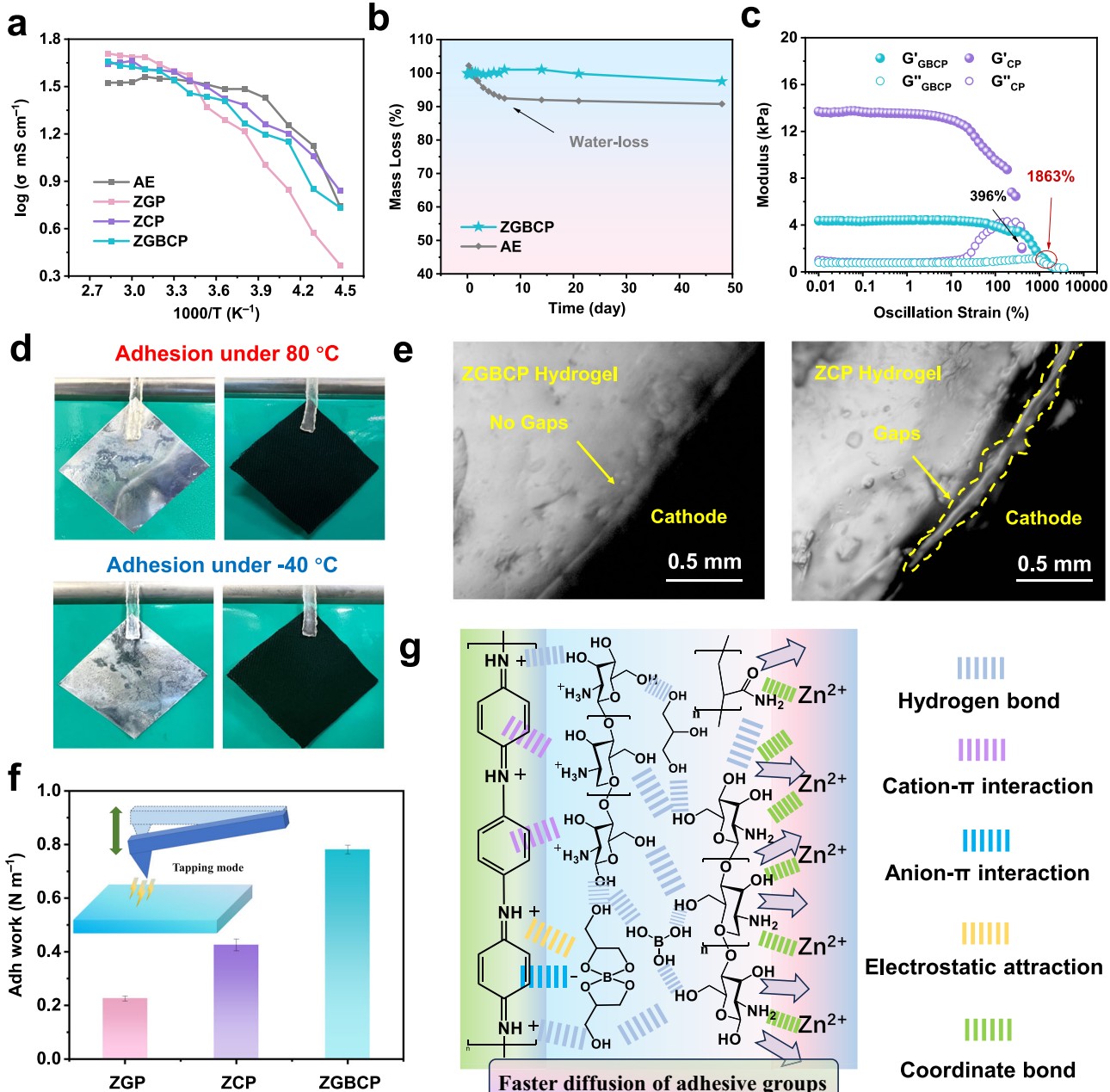

**Fig. 2 | The characterization of ion-conductive, mechanical, and thermal properties of different electrolytes. a** Ionic conductivities of four electrolytes at different temperatures from −50 to +80 °C. The difference between ZGBCP, ZCP, or ZGP electrolytes is that the ZCP or ZGP does not contain glycerol and boric acid or boric acid and chitosan. **b**, Water-retention capability of the ZGBCP and AE electrolyte. **c** The variation of storage modulus G' and loss modulus G" with strain oscillation of PAM/CS cross-linking network with/without the boric acid and glycerol. **d** Demonstration of ZGBCP electrolyte adhering to 5 × 5 cm² Zn anodes and PANI cathodes at different temperatures. **e** The optical images of the interface between the PANI cathode and ZGBCP or ZCP electrolyte. **f** Adhesion work tests of the ZGBCP, ZGP, and ZCP electrolytes (inset: schematic diagram of AFM tapping mode). **g** Schematic illustration of the adhesive mechanism of ZGBCP electrolyte.

when interacting with $BF_4^-$ anions than the $H_2O$ molecules with $BF_4^-$ anions (0.134; −0.16 eV). Thus, the function groups of −OH in glycerol, −OH in boric acid, $−CONH_2$ in PAM, and especially −OH and $−NH_2$ in CS will act as hydrogen bond donors to replace the water molecules around free $BF_4^-$, which can also confine the free water molecules and inhibit the water-induced vicious hydrolysis reaction of $BF_4^-$. Thereby, the reconstruction and protection of $BF_4^-$ structures can be realized (Fig. 3d, e and Supplementary Figs. 16, 17). Due to the reconstruction of both cationic and anionic solvation structures, the activation energy also decreases from 43.53 kJ mol⁻¹ for AE electrolyte to 27.02 kJ mol⁻¹ for ZGBCP electrolyte (Fig. 3f and Supplementary Fig. 18), which confirms the above hypothesis. Besides,

the transport properties obtained from MD simulations under different temperatures match up with the experimental results well (Supplementary Fig. 19). To elaborate on the role of glycerol in the solvation structures, the DFT calculation further revealed the asymmetric ESP because of the strong hydrogen bond between the central hydroxyl−H and one of the edge hydroxyl−O within the glycerol molecule as shown in the inset of Supplementary Fig. 20b[30–32]. Therefore, the desolvation zone and solvation zone are able to exit independently and convert rapidly thanks to the intramolecular hydrogen bond with short life, which guarantees the fast cationic solvation and desolvation process compared to other polyols (Supplementary Fig. 20)[33–35]. Besides, the stronger anionic affinity of boric

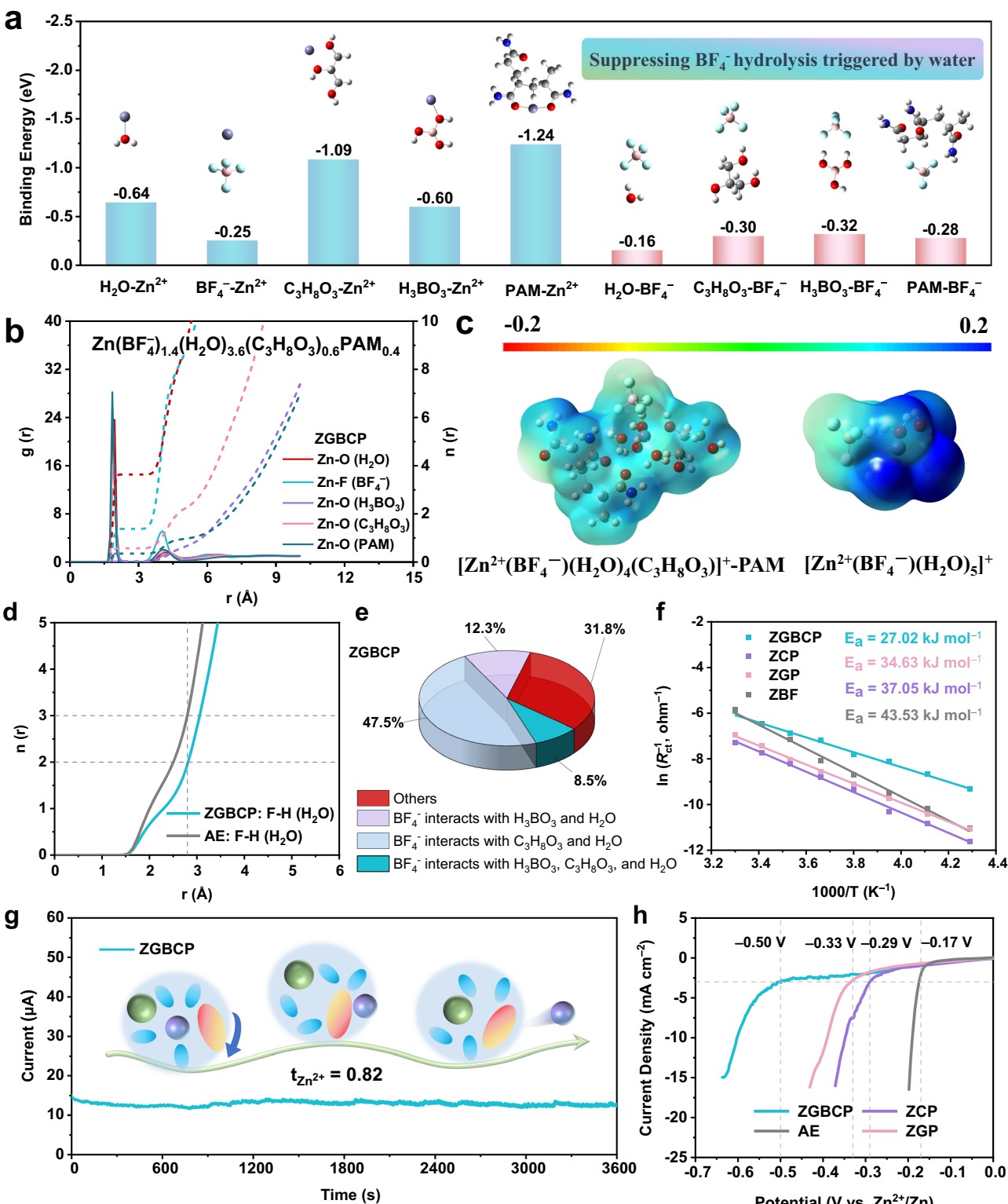

**Fig. 3 | Theoretical analysis and experimental characterization of the solvation structures and electrochemical stability of different electrolytes. a** Binding energy of $Zn^{2+}$ and $BF_4^-$ with species including $H_2O$, glycerol, boric acid, and PAM based on the B3LYP/6-311 + +g(d,p) level DFT calculations. **b** Cationic radial distribution function $g(r)$ and coordination number $n(r)$ different ligands. **c** The ESP distribution of cationic solvation structures based on the B97-3c level DFT calculations. **d** The $n(r)$ of F of $BF_4^-$ to the H of the $H_2O$ in the ZGBCP and AE electrolytes. **e** distribution of anions with different solvation configurations. **f** Activation energy $E_a$ based on the fitting of $\ln R_{ct}^{-1}$ versus $1000/T$ Arrhenius curves. **g** Transfer number test of the $Zn^{2+}$ ($t_{Zn^{2+}}$). **h** LSV curves of different electrolytes.

acid, glycerol, and CS is also conducive to the transfer process of cations, which interprets the high transfer number (0.82) and much lower desolvation energy barrier (3.46 eV) of $Zn^{2+}$ in the ZGBCP electrolyte as shown in above Fig. 3g and Supplementary Fig. 20f[36].

Thanks to the desolvation acceleration and suppression of free waters from attacking $BF_4^-$, the LSV tests showed a HER threshold potential expansion from −0.17 V for the AE to −0.50 V for the ZGBCP electrolyte. In contrast, the HER potentials for the ZCP and ZGP

electrolytes are only −0.29 and −0.33 V, respectively according to Fig. 3h. To further verify this phenomenon from a spectral version, a series of spectra tests were conducted. First, the ATR-FT-IR spectra revealed the tuned F⋯H−O−H interactions and stronger B−F bonds in the ZGBCP electrolyte compared to the other electrolytes (Supplementary Fig. 21). The elevated electron density in F sites of $BF_4^-$ anions with the addition of boric acid was also observed from the $^{19}$F NMR spectra (Supplementary Fig. 22). Besides, a lower HF content was found in the XPS F1$s$ results as shown in the Supplementary Fig. 23b, which are crucial for the interfacial stabilization of the Zn anodes. In conclusion, the CRACSS strategy is expected to improve the long-term cycling ability of $Zn(BF_4)_2$-based electrolytes towards AZMBs based on the above theoretical and experimental analysis.

## Investigations of interface behavior of Zn anode matched with different electrolytes

As analyzed above, the ZGBCP electrolyte with robust adhesion, high conductivity, and low HER risk was constructed by the CRACSS strategy successfully. To additionally verify the effectiveness from the perspectives of interfacial behavior, the HER tests of Zn anode with different electrolytes were characterized by in-situ optical microscopy (Supplementary Fig. 24). It is obvious that even in a quiescent state, vigorous gas evolution occurred at the Zn/AE interface due to the intense hydrolysis of $BF_4^-$. In contrast, no gas was observed for the Zn/ZGBCP interface, indicating that ZGBCP is capable of suppressing the HER side reaction. Furthermore, the ZGBCP electrolyte exhibited the fastest 3D diffusion behavior during the Zn plating process (Supplementary Fig. 25a), which benefits from the intimate interfacial adhesion and accelerated desolvation process as proved above. The smallest differential capacitance as illustrated in Supplementary Fig. 25b also confirmed the suppression of concentration polarization, which coincides with the results of the lowest adsorption/desorption, charge transfer, and diffusion impedance of the distribution of relaxation times (DRT) spectra of Zn∥Zn symmetric cells using ZGBCP electrolyte in Supplementary Fig. 25c.

From Fig. 4a, it can be seen that the symmetric cell based on ZGBCP electrolyte achieved stable cycling for over 3000 h at a current density of 1 mA cm$^{-2}$. In contrast, the cells using AE, ZGP and ZCP electrolytes occur short-circuiting after only about 100 h. In addition, optical profile images (Supplementary Fig. 26) and SEM images (Supplementary Fig. 27) of Zn anodes revealed the lower surface roughness ($R_a = 6.76$ μm) and the smoother surface without dendrites or side reaction products for the Zn electrode after cycling with ZGBCP electrolyte, suggesting its excellent compatibility with Zn anodes. On the other hand, for the Zn plating behavior, the XRD patterns of the cycled Zn anodes with different electrolytes were collected and shown in Fig. 4b. The cycled Zn anode with ZGBCP electrolyte exhibited a strong (002) preferential orientation with an I(002)/I(101) ratio as high as 2.20, indicating ZGBCP electrolyte can induce the preferential plating behavior and alleviate the dendritic short-circuiting, which is consistent with the previous reports that also use solvation structure regulation strategies[37, 38]. Furthermore, Tafel tests, as illustrated in Supplementary Fig. 25d, prove that the ZGBCP electrolyte exhibited exchange current densities nearly comparable to AE electrolyte, demonstrating the ZGBCP electrolyte offers ultra-fast stripping/plating reaction kinetics. As a result, when using the ZGBCP electrolyte, the polarization voltage of the Zn∥Zn symmetric cell (Fig. 4c) is lower than that of the cell with other electrolytes. Moreover, even at a high current density of 10 mA cm$^{-2}$, the polarization voltage is only 215 mV.

Furthermore, the ZGBCP electrolyte can alleviate the corrosion reaction, which is attributed to the reconstruction of the anionic solvation structure and the capturing of the free F$^-$ anions by the glycerol and boric acid (Fig. 4d). Thus, the ATR-FT-IR spectra of the Zn anode interface after cycling with ZGBCP electrolyte shows a weaker Zn−F

stretching vibration signal and a stronger B−F stretching vibration signal, indicating less transformation of $BF_4^-$ into $ZnF_2$ (Fig. 4e). In addition, unlike the uneven corrosion layer on the Zn anodes surface after cycling with the AE electrolyte, the SEI at the Zn/ZGBCP interface is denser and has a more uniform F distribution of ~5 nm (Supplementary Figs. 28 and 29). To further elucidate the mechanism of SEI formation promoted by ZGBCP electrolyte and its structure-performance relationship, XPS tests were conducted at different depths of the Zn electrode etched by Ar$^+$ beams. As seen in Fig. 4f and Supplementary Figs. 30, 31, a thinner and more homogenous SEI is generated on the Zn electrode after cycling using the ZGBCP electrolyte and exhibits lower fluorine and higher zinc content compared to the SEI that lacks $Zn^{2+}$ for the Zn electrode cycled in the AE electrolyte. As a result, the optimized $Zn^{2+}$-rich SEI can provide a shorter solid-phase mass transport path and avoid concentration polarization on the basis of protecting Zn electrodes (Fig. 4g).

In order to understand the balance between the kinetics and anti-corrosion of Zn stripping/plating process endowed by the ZGBCP electrolyte and clarify the different roles of glycerol, boric acid and chitosan in the electrolyte, Tafel tests of Zn electrode paired with different electrolytes were conducted under different temperatures from −40 to +50 °C (Supplementary Fig. 32). Among all the electrolytes, AE electrolyte showed the most rampant corrosion and HER reactions at +50 °C. Furthermore, the most balanced level of exchange current density along with equilibrium potential at −40 °C was observed using ZGBCP electrolyte thanks to the lowest energy barrier of desolvation, optimized SEI, and robust adhesion. The DRT spectra of the Zn∥Zn symmetric cells using AE and ZGBCP electrolyte further corroborate this conclusion, since the diffusion and charge transfer impedance increase more dramatically along with the decrease of temperature for the AE electrolyte (Fig. 4h). It needs to point out that the polarization voltage can be further decreased to 190 mV at 10 mA cm$^{-2}$ (Supplementary Fig. 33) and is as low as 280 mV at ultralow temperature of −40 °C (Supplementary Fig. 34) when using an electroplated Zn electrode, which is the record-low value among all the reported electrolytes using $Zn(BF_4)_2$ salt (Fig. 4i)[4, 15, 16, 37, 39–44]. In summary, the perfect balance of stripping/plating kinetics and interfacial SEI endowed by the ZGBCP electrolyte indicate the five components of glycerol, boric acid, CS, PAM, and $Zn(BF_4)_2$ in the electrolyte play synergistic roles in regulating solvation structures and decreasing polarization. This paves the way for ultra-stable AZMBs even under a wide working temperature range.

## The electrochemical performance of the full cells with different electrolytes

As mentioned above, the CRACSS strategy effectively regulates both the cationic and anionic solvation structures. To verify the influences of the CRACSS strategy on electrochemical performance, polyaniline (PANI), which can store both anions ($BF_4^-$) and cations ($Zn^{2+}$), was selected as the cathode to investigate the compatibility and effect of ZGBCP electrolyte. Impressively, as seen from Fig. 5a, the Zn|ZGBCP| PANI full cell exhibited better rate performance, and a reversible specific capacity exceeding 140 mAh g$^{-1}$ was achieved at a current density of 0.2 A g$^{-1}$ (Fig. 5b). Even at a high current density of 20 A g$^{-1}$ (corresponding to 35 mA cm$^{-2}$), the specific capacity remained as high as 120 mAh g$^{-1}$. The full cell with AE electrolyte showed fast capacity decay upon increasing rates, while the coulombic efficiency (CE) and energy efficiency (EE) of the full cells were well maintained when using the ZGBCP as the electrolyte (Supplementary Fig. 35a, b). In particular, a comprehensive comparison between the reported Zn∥PANI batteries with this work reveals that the Zn|ZGBCP|PANI full cell possesses higher energy and power density as illustrated in Supplementary Fig. 35c. In addition, the full cell with ZGBCP electrolyte demonstrated ultralong cycling stability of over 10,000 cycles at a high rate of 15 A g$^{-1}$ (Fig. 5c) and almost same polarization and diffusion coefficient as the

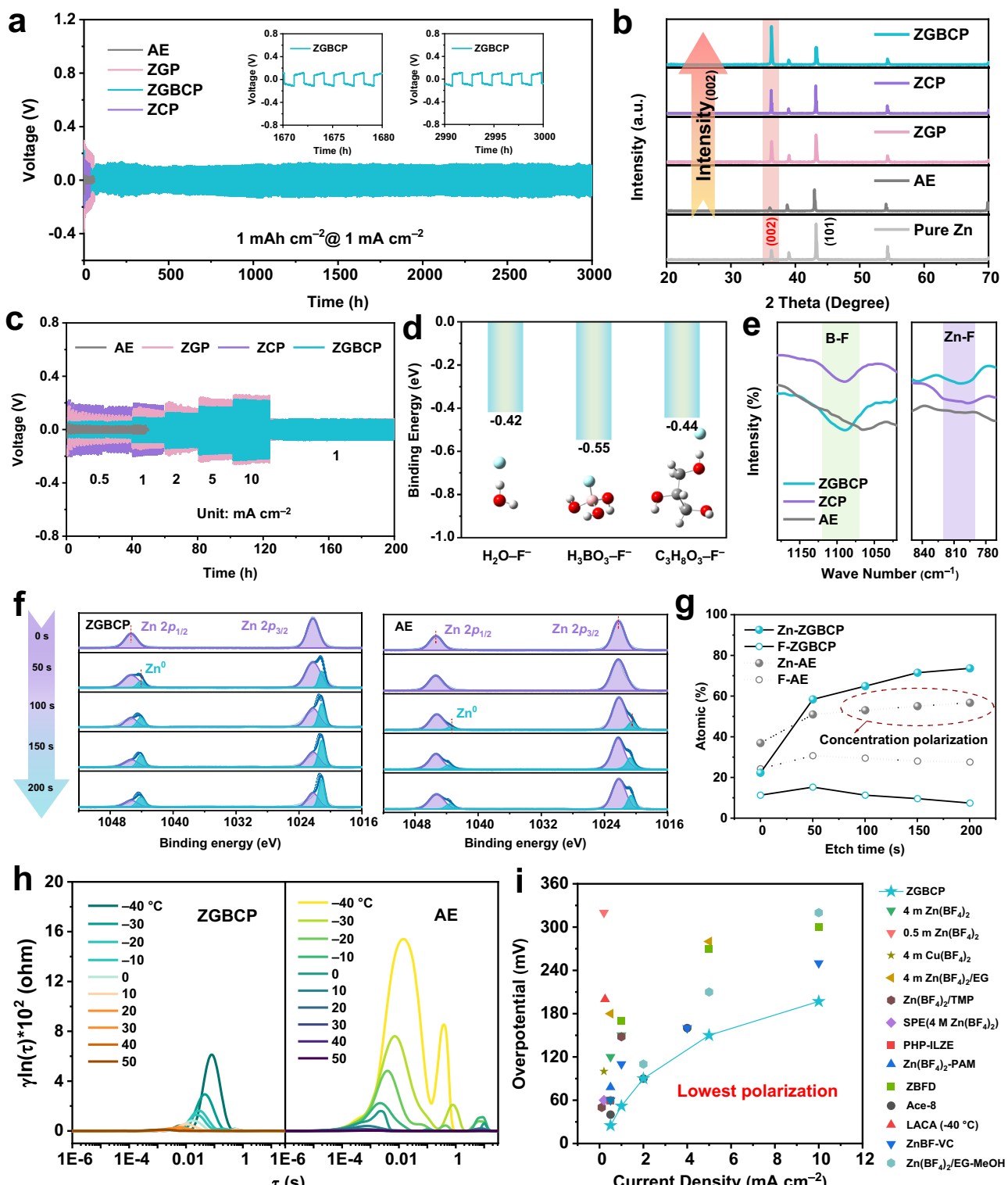

**Fig. 4 | Interfacial stripping/plating reaction kinetics and interfacial SEI of Zn anode with different electrolytes. a** Cycling stability of the Zn∥Zn symmetric cell (1 mAh cm⁻², 1 mA cm⁻²). **b** XRD spectra of Zn anodes using different electrolytes after 100 cycles. **c** Rate performance tests of Zn∥Zn symmetric cells with different electrolytes. **d** Binding energy of between F⁻ with $H_2O$, boric acid, and glycerol molecules based on the B3LYP/6-311 ++ g(d,p) level DFT calculations. **e**, ATR-FT-IR spectra at the Zn anodes surface using ZGBCP, ZCP, and AE electrolytes. **f** XPS spectra of Zn $2p$ of Zn anodes at different depths. **g** The profiles of the atomic percentage of Zn and F atoms versus etching time. **h** DRT spectra of the Zn∥Zn symmetric cells tested at different temperatures from −40 to +50 °C. **i** Comparison of polarization voltages of Zn∥Zn symmetric cells at different current densities of reported Zn(BF₄)₂-based electrolytes and this work[4,15,16,37,39–44].

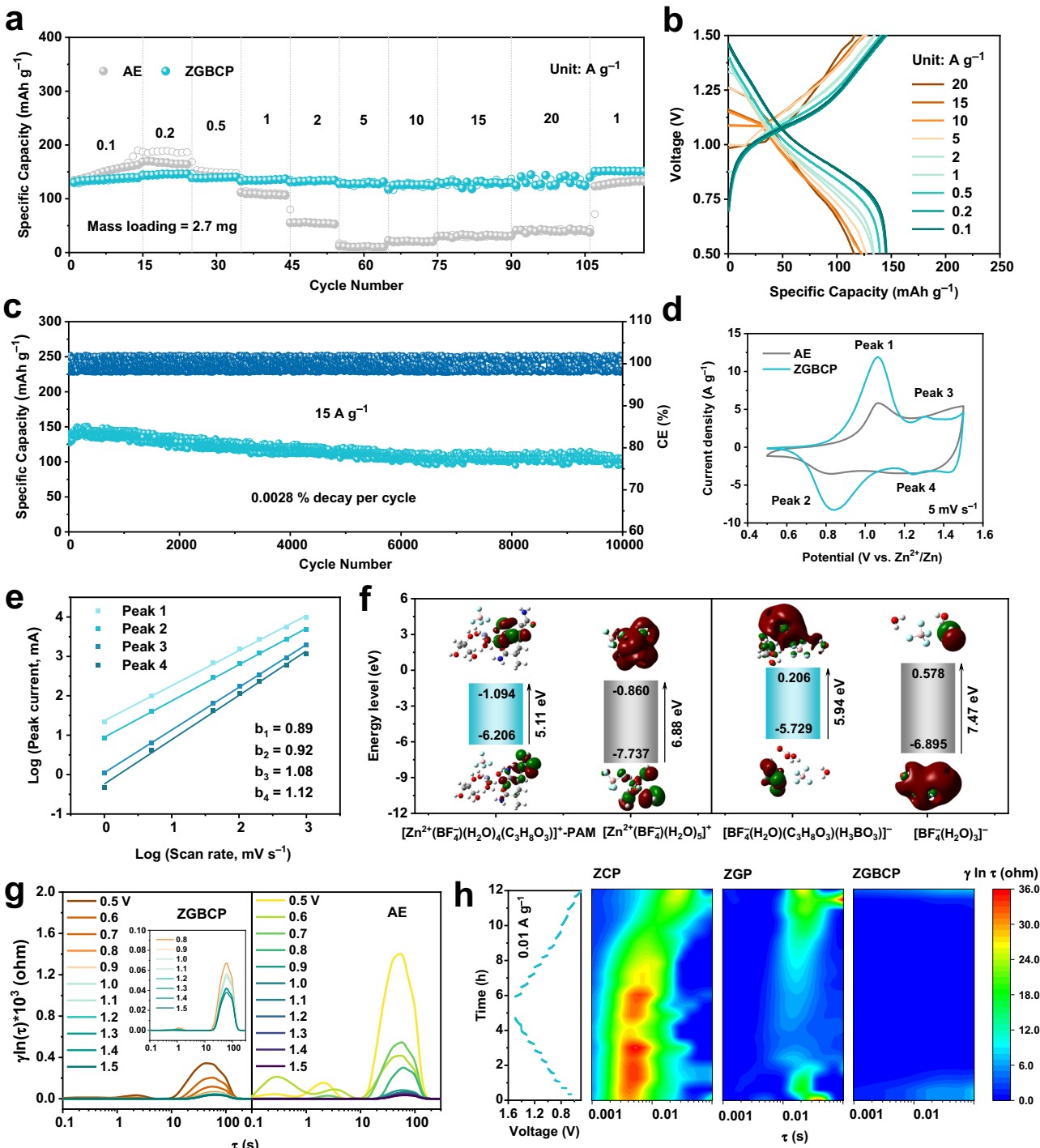

**Fig. 5 | Electrochemical performance and reaction mechanism analysis of Zn|| PANI full cells with different electrolytes. a** Rate performance of full cells. **b** Charge and discharge profiles of the full cell using ZGBCP electrolyte at different current densities. **c** Cyclic performance of the full cell using ZGBCP electrolyte at 15 A g⁻¹. **d** CV curves of full batteries using AE and ZGBCP electrolytes. **e** log ($i_p$)

versus log ($v$) profile. **f** Energy level of the representative cationic (left) and anionic (right) solvation structures from AE and ZGBCP electrolytes based on the B97-3c level DFT calculations. **g** In-situ DRT spectra of the full cells using the ZGBCP and AE electrolytes at different voltages. **h** In-situ DRT spectra of full cells using different electrolytes.

liquid cell (Supplementary Fig. 35d). Moreover, for conversion-type electrode reactions based on halides shown in Supplementary Fig. 36, the ZGBCP electrolyte also renders the cathode excellent reversibility. Even at a high current density of 15 A g⁻¹, the Zn|ZGBCP|I₂ full cell provided a specific capacity of 124 mAh g⁻¹ after 12,000 cycles, demonstrating the universality of the ZGBCP electrolyte.

In order to clarify the underlying mechanism of improved kinetics by the ZGBCP electrolyte, a series of electrochemical and

spectroscopic analyses was conducted. The cyclic voltammetry (CV) curves of Zn|ZGBCP|PANI full cells using different electrolytes were compared in Fig. 5d and Supplementary Fig. 37. The full cell with ZGBCP as the electrolyte shows the most well-defined, lowest polarization, and highest anodic/cathodic peak current. Moreover, the main cathodic peak potential barely shifted as the scan rate gradually increased from 1 to 10 mV s⁻¹ as illustrated in Fig. 5e and Supplementary Fig. 38, indicating that ZGBCP

significantly enhances the reaction kinetics of the PANI cathode. Based on the Eq. (3):

$$i = kv^b \tag{3}$$

the peak current–peak potential relationship was linearly fitted, yielding a $b$ value close to 1, indicating the surface capacitive control of the anionic/cationic coordination reaction, which is consist with the dominant pseudocapacitive contribution.

Thorough DFT calculations were further conducted to analyze the solvation structures of cations and free anions. As seen from Fig. 5f and Supplementary Fig. 39, there are dramatic reductions in the energy gap exceeding 1–2 eV for all the cationic solvation structures in the ZGBCP electrolyte. The elevated active electron states can further facilitate (de)solvation and accelerate charge transfer during the charge and discharge process, enhancing the electrochemical reactivity[45]. Besides, the elevated HOMO levels of the anionic solvation structures dominated by other ligands instead of $H_2O$ are beneficial for the charging process of PANI, which can also alleviate the irreversible water and proton consumption during the cycling as seen in Supplementary Fig. 40. To further analyze the reaction kinetics of the cathode, in-situ EIS tests were conducted using different electrolytes, and the corresponding DRT spectra were obtained as shown in Fig. 5g, h. Remarkably, the ZGBCP electrolyte showed almost no charge transfer impedance in the cathode side from the range of 0.1 -10 s and a much lower diffusion impedance compared to the AE electrolyte according to Fig. 5g, which is consistent with the lower polarization observed in the CV curves between Peaks 1 and 2. On the other hand, due to the superior interface adhesion macroscopically, optimized SEI microscopically, and lower desolvation energy barrier in the molecular scale tuned by the CRACSS strategy, the full cell using ZGBCP electrolyte exhibits almost no charge transfer impedance on the Zn anode side during the whole charge and discharge process in comparison with AE. At room and elevated temperature, it can be seen from Supplementary Fig. 41 that the chain structures of PANI cathode decompose faster with ZCP electrolyte, which is proved by the N 1$s$ XPS, UV–Vis, and RAMAN spectrums. In contrast, the structures of the PANI cathode are well maintained when using ZGBCP electrolyte, indicating its capability of enhancing the thermal stability of the PANI cathode.

Since the ZGBCP is stable in a wide temperature, the Zn|ZGBCP|PANI full cell was then evaluated at various temperatures. As seen from Fig. 6a, the cell can stably operate from −50 to +100 °C and deliver decent specific capacities of 103.1 and 100.7 mAh g$^{-1}$ at −50 and +100 °C, respectively. Furthermore, whether at low temperatures (−40 and −50 °C) or elevated temperatures (+50 °C), the ZGBCP-based full cells all exhibit excellent rate capability (Fig. 6b and Supplementary Fig. 42), and the specific capacity can well-recover under the transition of low temperatures and elevated temperatures. Such a wide working temperature is superior to the so-far reported hydrogel-based AZMBs, as shown in Fig. 6e[27, 42, 46–59], indicating the most advanced cell performance rendered by the ZGBCP electrolyte. As mentioned, the PANI suffers from thermal decomposition and even HER at elevated temperatures, which can be seen from the CV and GCD curves using AE in Supplementary Fig. 43. However, when matched with ZGBCP electrolyte, it can stably cycle for 300 cycles even at +80 °C, suggesting ZGBCP is indeed capable of inhibiting the high-temperature decomposition as shown in Supplementary Fig. 43c.

As to long cycling stability, the full cell shows more impressive performance. At −40 and +50 °C, cycle life up to 30,000 cycles and 5000 cycles with a capacity retention rate of 85.4% and 91.3%, respectively, can be achieved (Fig. 6c, d). Moreover, the specific capacity is even higher than 200 mAh g$^{-1}$ at +50 °C due to the thermal activation and inhibited HER by the CRACSS strategy. Intriguingly, a wearable electronic device can be powered easily by three Zn|ZGBCP|PANI pouch-cells in series even at −40 and 70 °C, as displayed in Fig. 6f

and Supplementary Fig. 44, which proves the effectiveness and practicality of the CRACSS strategy. Different from the various reported works aiming at AZMBs with wide temperature range through replacing anti-freezing solvent[46, 47, 50, 51, 53–55], tuning concentration of chaotropic salts[42, 49, 58], improving electrodes adhesion[27, 52, 57], and modifying ions transport behavior[56, 59], the CRACSS strategy is a state-of-art approach with combining properties (Fig. 6e). In conclusion, based on the dual cross-linking framework with robust adhesion and faster motion of chain segments as designed, the reconstruction of the cationic solvation structure enables the ZGBCP electrolyte to suppress the vicious HER and accelerate the stripping/plating kinetics of Zn anodes. Furthermore, the anionic solvation structures are reconstructed to prevent the corrosion induced by the $BF_4^-$ decomposition and accelerate the charge transfer process towards cathodes, enabling the ZGBCP electrolytes for ultra-stable and ultrawide-temperature-range AZMBs (Fig. 6g).

## Discussion
The strong solvation effect by $H_2O$ molecules on zinc ions contributes to undesirable side reactions in AZMBs, such as hydrogen evolution, passivation, and dendrite formation. The influence of anions' solvation structures on Zn anodes remains relatively unexplored for regulation. Herein, through the co-reconstruction of both the cationic and anionic solvation structures inspired by the boron-polyol chemistry, the desolvation and charge transfer process of cationic solvation structures accelerated and the hydrolysis of free anions is suppressed, rendering the ZGBCP electrolyte with robust adhesion to electrodes, lower HER and corrosion tendency, and faster reaction kinetics during cycling under extreme temperatures. Based on these, the ZGBCP electrolyte renders the AZMBs ultrawide working temperature (−50 to +100 °C) and ultralong cycle life (30,000 cycles), which are the state-of-the-art performance in the hydrogel-based AZMBs, which further validates the feasibility of the CRACSS strategy and provides a new perspective for the development of electrolytes towards wide-temperature range AZMBs.

## Methods
### Preparation of ZGBCP, ZCP, ZGP electrolytes
**ZGBCP electrolyte.** First, 2 g glycerol (AR, Aladdin) and 3 g deionized water were mixed at 80 °C. After the mixture was evenly mixed, 0.6 g boric acid (AR, Sinopharm) and 0.085 g deacetylated chitosan (DAC > 95%, bidepharm) was sequentially added and further stirred to fully dissolve without insoluble floccule at 80 °C. Second, 2 M Zn(BF$_4$)$_2$ (AR, Aladdin) aqueous solution of 10 mL was added to obtain a clear and transparent solution. After that, 2 g acrylamide (AR, Aladdin) monomers together with 0.02 g MBAA (AR, Alfa Aesar) as crosslinking agent and 0.02 g Irgacure 2959 (AR, BASF) as photo-initiator were added to the above solution. Finally, the gelation process was conducted under UV light for 20 min in a mold to obtain the expected ZGBCP hydrogel electrolyte.

**ZCP electrolyte.** Firstly, 0.75 g of glacial acetic acid (AR, Aladdin) and 4.25 g of deionized water were mixed before 0.085 g of deacetylated chitosan was added. The subsequent preparation process was the same as that of the above ZCBGP hydrogel.

**ZGP electrolyte.** The preparation of the ZGP electrolyte was the same as that of the ZGBCP one except that the ZGP electrolyte was prepared without boric acid and deacetylated chitosan.

### Synthesis of PANI cathode and $I_2$ cathode
The PANI cathode was prepared through a typical in-situ polymerization strategy[60]. Specifically, the hydrophilic carbon cloth (CC) was first immersed sufficiently in the solution containing 1 M HCl and 3.65 mL of aniline (AR, Aladdin) monomers in an ice water bath. After that, the

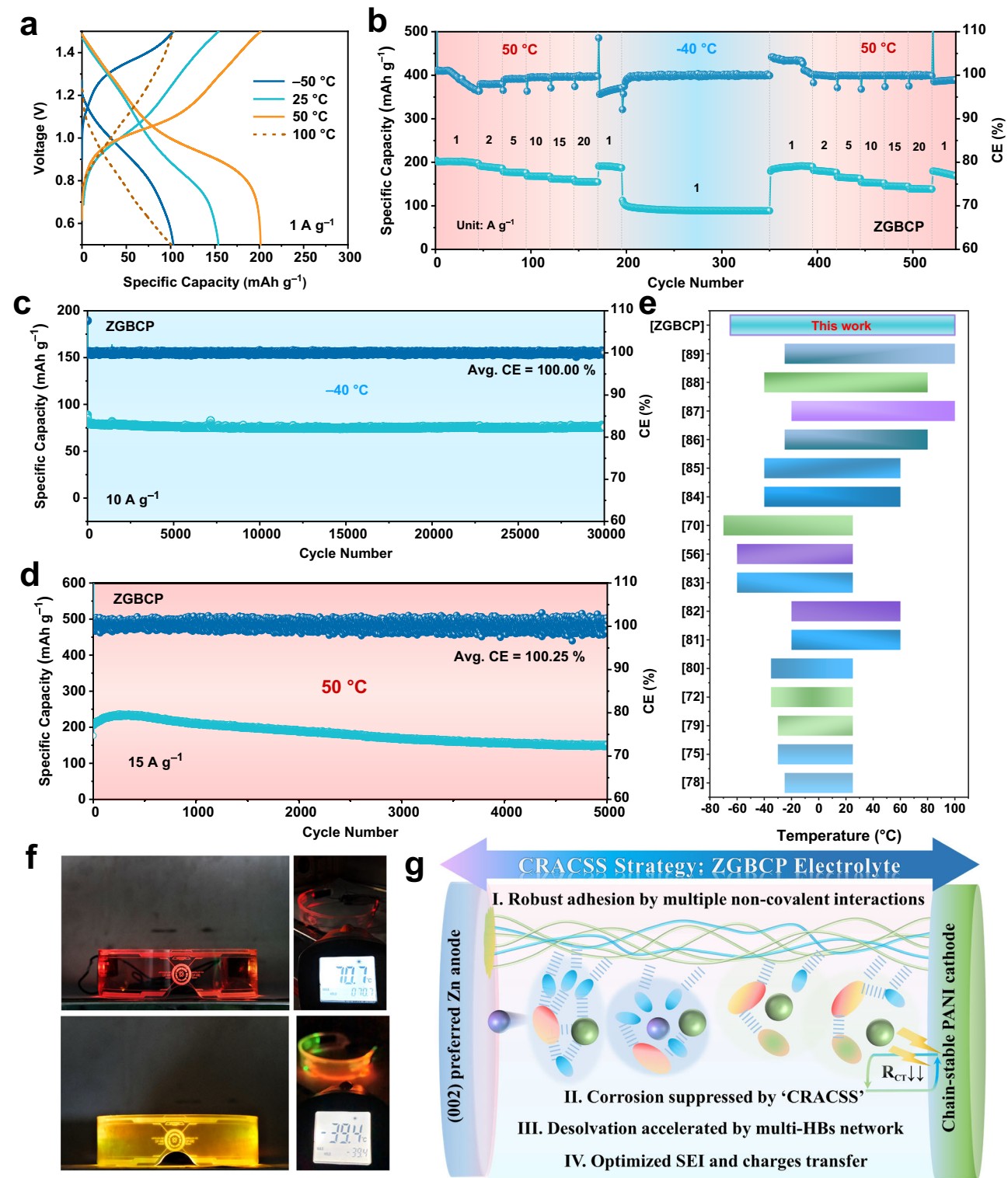

**Fig. 6 | Electrochemical performance of Zn|ZGBCP|PANI full cells at extreme temperatures. a** Charge and discharge profiles under different temperatures from −50 to 100 °C at 1 A g⁻¹. **b** Cycling performance at −40/50 °C. **c** Long-term cycling performance at −40 °C at 10 A g⁻¹. **d** Long-term cycling performance at 50 °C at 15 A g⁻¹. **e** Comparison of the temperature windows of the reported AZMBs with different electrolytes[27, 42, 46–59]. **f** Demonstration of electronic devices powered by three pouch-cells in series at −40 and 70 °C. **g** Summary of the regulation mechanism based on the CRACSS strategy in the full cells.

solution of 2.28 g ammonium peroxydisulfate (APS, AR, Aladdin) as the oxidant dissolved in 1 M HCl was dropped into the above solution to promote the polymerization reaction. After 3 h, the dark-green PANI@CC cathodes with free-standing structure were washed with 1 M HCl for three times and dried in the dark conditions for 12 h in an

atmospheric environment. Finally, the PANI@CC electrode was cut into a disk with a diameter of 14 mm and areal mass loading of 1–2 mg cm⁻².

The I₂ cathode was prepared by the gas-adsorption method. In detail, active carbon black (AC, YP-50F, Kuraray) and iodine (AR,

Aladdin, weight ratio: 1:3) were homogeneously mixed and then heated at 120 °C for 12 h in vacuum-sealed ampule. Then the $I_2$@AC composite powders were collected after heating to 60 °C for 5 h to remove excess iodine. The $I_2$@AC cathode was then prepared by mixing 80 wt% of $I_2$@AC, 10 wt% of Super P as a conductive agent, and 10 wt.% of CMC as the binder in water solvent and cast onto a clean carbon cloth current collector with a diameter of 14 mm and areal mass loading of 1.7 mg cm$^{-2}$. The iodine content in the final cathode was 25 wt%.

### Preparation of electroplated Zn anode
To prepare the electroplating solution, mixture of 18 g $Zn(BF_4)_2$, 2.5 g $H_3BO_3$, and 11 g $Zn(CH_3COO)_2$ (AR, Aladdin) in deionized water, glycerol and anhydrous ethanol (volume ratio: 8:1:1). A current density of 40 mA cm$^{-2}$ was conducted for 1 h to plate Zn on the hydrophilic carbon cloth.

### Materials characterization and electrochemical measurement
**Materials characterization.** The shear adhesion test of the hydrogel electrolytes was conducted on a universal material testing machine (Instron-5592) with the unified standard specimens (adhesive area: 5 × 3 cm$^2$, thickness: 300 μm). The pH of the boric acid and glycerol solution was measured using a pH analyzer (METTLER TOLEDO, S210). The rheological properties of the hydrogel electrolytes were measured by rheometer (HAAKE MARS60) at a fixed frequency of 10 Hz. The thermal properties of hydrogels were characterized by a thermogravimetric analyzer (TGA, NETZSC, STA 409 PC) with a heating rate of 5 °C min$^{-1}$ in the air atmosphere with $N_2$ as the protecting gas. The electronic structure and corresponding composition were characterized by X-ray photoelectron spectroscopy (XPS, Thermo Scientific K-Alpha). For the treatment of cycled Zn electrodes, the gel residues adhering to the surface of the cycled zinc foil were completely peeled off with great care. Given the high solubility of $Zn(BF_4)_2$ in ethanol and the relatively low boiling point of ethanol, the zinc foil cycled with ZGBCP and AE electrolytes were then rinsed with anhydrous ethanol to prevent any potential influence of secondary hydrolysis of $BF_4^-$ on the determination of F species in the SEI. Subsequently, the samples were vacuum dried, transferred under inert atmosphere protection, and promptly subjected to relevant tests to avoid secondary oxidation and radiation damage. Characteristic functional groups and bond structures were analyzed by the attenuated total reflection Fourier transform infrared spectroscopy (ATR-FTIR, Thermo Scientific Nicolet iS5) in the range of 4000–400 cm$^{-1}$ with a resolution of 4 cm$^{-1}$. The $^{19}F$ NMR spectra were characterized using the heavy water as a deuterated reagent (Bruker Avance NEO 400 MHz). The phase structures of Zn anodes before and after cycling were identified by X-ray diffraction (XRD, Rigaku Ultima diffractometer with Cu Kα radiation, $\lambda = 1.5418$ Å). The morphology and roughness of the Zn anodes were analyzed by an optical profiler (Mahr LD130). Scanning electron microscopy (SEM, JEOL, JSM-6510) equipped with the energy dispersion spectrometer (EDS) was employed to observe the morphology and element distribution of the Zn anodes. To analyze the structural evolution of the PANI cathode, the UV-Vis spectra were characterized by the UV/Visible/Near Infrared Diffuse Reflectometer (Shimadzu UV-3600) through the Integral sphere mode.

**Electrochemical measurement.** Electrochemical impedance spectroscopy (EIS), chronoamperometry (CA), linear sweeping voltammetry (LSV), and cyclic voltammetry (CV) were performed on the electrochemical workstation (Autolab instrument PGSTAT302N). The AC signal ranging from 0.01 Hz to 1000 kHz was used for all the EIS tests. Galvanostatic charge/discharge (GCD) cycling tests were carried out on the LAND CT2100A. Tafel and LSV tests were conducted using a scanning rate of 0.5 mV s$^{-1}$ with Zn as the working electrode, Zn as the counter electrode, and another Zn as the reference electrode. The DRT analysis was performed by using DRT

Tools[61]. Symmetric cells were assembled using commercial Zn foils (100 μm) and electroplated Zn electrodes. As to the full cells, an electroplated Zn electrode was used as an anode, and PANI or $I_2$ was used as a cathode. Pouch cells were fabricated by clamping the ZGBCP electrolyte in the middle of the electroplated Zn electrode and PANI cathode with an area of 5 × 5 cm$^2$. A packaging bag (6 × 6 cm$^2$) was used to accommodate the cell through hot-pressing, and stainless-steel sheets as collectors.

The transfer number of zinc ions ($t_{Zn^{2+}}$) was calculated according to the CA test using the equation:

$$t_{Zn^{2+}} = \frac{I_s(\Delta V - I_0 R_0)}{I_0(\Delta V - I_s R_s)}$$

where $I_0$ and $I_s$ are the initial and steady-state current, $\Delta V$ is the applied constant potential (10 mV), and $R_0$ and $R_s$ are the initial and steady-state interface resistance.

The differential capacitance curve was calculated from the equation:

$$C = -(\omega Z_{im})^{-1}$$

where $C$ is the differential capacitance and $\omega$ is the angular frequency, $Z_{im}$ is the imaginary part of the impedance, and 1000 Hz was selected as the specific frequency.

In addition, in order to explore the diffusion behavior of the full cells using different electrolytes, a GITT test was conducted, and the diffusion coefficient was calculated based on the equation:

$$D = \frac{4}{\pi \tau} L^2 \left(\frac{\Delta E_s}{\Delta E_t}\right)^2$$

where $L$ is the electrode thickness, $\tau$ is the constant current pulse time, $\Delta E_t$ is the voltage change caused by charging and discharging at 0.5 A g$^{-1}$ (40 s), and $\Delta E_S$ is the voltage change caused by pulse (1800 s).

**Computational methods.** MD simulation was executed by GROMACS, and the generalized amber force field (GAFF) parameters were used in the MD process[62]. Packmol was applied to generate initial coordinates for MD[63]. The Sobtop code was used to generate the necessary force field parameters for the simulation systems[64]. The partial charges on atoms were obtained using the restrained electrostatic potential (RESP) method and calculated with Multiwfn software[65].

Before starting the MD simulation, the initial configurations were relaxed using a conjugate gradient minimization scheme. The step size for the conjugate gradient minimization scheme was set to 0.01 nm, and the cycle was set to 10,000 steps. The minimization force was considered converged when it was <50 kJ mol$^{-1}$ nm$^{-1}$. Van der Waals interaction was calculated using the cut-off method, while atomic electrostatic interaction was calculated by particle mesh Ewald (PME) with both the cut-off and PME distances set to 1.0 nm[66]. The simulation setup parameters were determined by the experimentally quantified component ratios to reflect the intrinsic state of the different electrolytes.

The system was then equilibrated with a pressure of 1.0 bar to attain the desired density using the Nose–Hoover method for temperature control at 300 K. The time constant was 0.1 ps. Equilibrium was performed for 10 ns for all systems with a time-step of 0.001 ps. Finally, production was run for 200 ns, with pressure control switched to the Parrinello–Rahman method, the time constant was 1.0 ps, and the compressibility was 4.5 × 10$^{-5}$ bar$^{-1}$. To impose constraints on hydrogen bonds, the linear constrain solver (LINCS) algorithm was used[67]. The visualization and analysis of MD results were conducted by the VMD software[68].

The binding energies were calculated using the Gaussian (G09) program at the B3LYP-D3/6-311 ++g(d, p) level[69–71]. The implicit universal solvation model based on solute electron density (SMD) with a dielectric constant of water was employed to investigate the influence of the solvents[72]. The binding energy is calculated based on the following formula:

$$E_{be} = E_{A+B} - (E_A + E_B)$$

Among them, the $E_{A+B}$ is the Gibbs free energy of combination A and B and the second item is the sum of the Gibbs free energy of individual A and B. The solvation structures were calculated at the B97-3c level by the ORCA 5.01 package[73, 74]. The bond order and electrostatic interaction energy were processed by the Multiwfn software[65, 75].

## Reporting summary
Further information on research design is available in the Nature Portfolio Reporting Summary linked to this article.

## Data availability
The data that support the findings of this study are available within the text, including the Methods, and Supplementary information. Raw datasets related to the current work are available from the corresponding author on request.

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

## Acknowledgements

This work was supported by the National Key R&D Program of China (2024YFE0101100). This work was also financially supported and sponsored by the National Natural Science Foundation of China (Grant No. 52171147), Ten-thousand Talents Program, K.C. Wong Pioneer Talent Program, and Shanghai Pujiang Program (Grant No. 19PJ1410600). The authors also thank Shiyanjia LAB (www.shiyanjia.com) for supporting spectral tests.

## Author contributions

Y.L. and X.C. supervised the research. L.Y. conceived, carried, collected, and analyzed the experimental and simulating data. J.L. offered suggestions about simulation. B.W. offered suggestions about experiments. Y.L., F.Z., and J.L. wrote the paper. All authors participated in discussions of the research.

## Competing interests

The authors declare no competing interests.
