## [Peer Review File · Nature Communications]

Reconstruction of Zinc-metal battery solvation structures operating from -50~+100 •REVIEWER COMMENTS

Reviewer #1 (Remarks to the Author):

This work utilizes a multi-component electrolyte with $\text{Zn}(\text{BF}_4)_2$ -glycerol-boric acid-chitosan-polyacrylamide and achieves good performance over a wide temperature range of -50 to 100 °C. A new solvation view is proposed, highlighting reconstructing the anionic and cationic solvation structures, which contribute to high ionic conductivity and cation transfer numbers. The proposed structure is interesting, but comparisons with similar works in gel or liquid electrolytes are lacking. The complexity of the electrolyte composition (with over 3 components except salt) may lead to results or discussions that do not fully explain the real case. Here are some concerns and suggestions for improvement:

1. The claim in sentence 80 about the robust interphase of the ZGBCP electrolyte and the dual crosslinking network involving esterification, protonation, and polymerization reactions needs further explanation to clarify how these reactions contribute to the function.
2. The reconstruction of cationic and anionic solvation structures is emphasized but not clearly summarized in the paper.
3. The statement about the strong hydrogen bond between central hydroxyl-H and one of the edge hydroxyl-O within the glycerol molecule contributing to rapid desolvation lacks evidence to support it.
4. The reliability of transfer number measurements is questioned due to potential influences from electrode surfaces and applied voltages. More evidence or alternative methods, such as calculating desolvation energy of cations, should be provided.
5. The differences in solvation structures of cations and anions compared to other reported electrolytes are not clearly demonstrated in the current simulation results. The reconstruction of anion solvation structures should be more clearly presented.

Reviewer #2 (Remarks to the Author):

The paper titled "Co-reconstruction of anionic and cationic solvation structures for ultra-stable operating of zinc-metal batteries under $-50\sim+100$ oC" by authors L. Yao, J. Liu, F. Zhang, B. Wen, X. Chi and Y. Liu deals with the regulation of cationic/anionic solvation structures in aqueous zinc metal batteries by using multicomponent electrolyte, consisted of $\text{Zn}(\text{BF}_4)_2$, glycerol, boric acid, chitosan and polyacrylamide (ZGBCP). The authors presented a detailed experimental and computational investigation of solvation structure and a comprehensive understanding of the mechanism in proposed battery devices. Also, the performances of proposed water-based zinc batteries are promising. All the experiments are well-thought-out and well-designed resulting in a coherent story even though the investigated system is complicated. Overall, the article is interesting, actual, and novel.

Recommendation: Major revision

Below are some comments, hope will help to modify this paper:

Major remarks:

1)DFT calculations. The use of DFT calculations to explain the coordination of Zn^{2+} and BF_4^- in terms of binding energies and bond order is a great idea.

Is there any particular reason why chitosan is excluded from the discussion and calculation of binding energies? From Scheme 1g, chitosan is proposed to be one of the coordination sites for Zn-ion, along with acrylate, which is a reasonable assumption due to the existence of $-\text{OH}$ and $-\text{NH}$ groups in the chitosan structure. However, there is no data for Zn^{2+} -chitosan binding energies from DFT calculation which is contradictive. Also, does BF_4^- have any interactions with chitosane or PAM based on DFT

calculations? Please, revise the DFT models and change the proposed solvation model accordingly.

2) MD simulations. It is difficult to analyze presented RDFs data from MD simulations and to evaluate the validity of the used force field.

Authors should provide the number of molecules (in Supporting information) and the size of the simulation box used in MD simulations during the production run. Did the authors perform any validation of the used force field? I will strongly recommend showing the obtained data in the form of coordination number since presented RDFs (Figure 2e) are impossible to analyze and therefore the corresponding discussion is confusing. Also, please note that chitosan is again missing like in the DFT calculations, so the proposed discussion is ambiguous and needs to be revised accordingly.

3) Please, provide the procedure for treatment of the electrode before post-mortem analysis. SEI formation could be easily influenced by post-treatment of the electrode, and it is necessary to include a detailed handling procedure of the electrode.

Minor remarks:

1. XPS data. It would be interesting to include a wide range of data for XPS measurements in Supporting information. The shift in XPS data is obvious (Figure 3e and Fig. S14). Did the authors perform any internal calibration of XPS data?
2. English should be improved in the entire Manuscript.
3. Chapter before "Methods" should be Conclusion, not Discussion

Reviewer #3 (Remarks to the Author):

The manuscript by Yao et al introduces a novel quinary electrolyte of $\text{Zn}(\text{BF}_4)_2$ / glycerol / boric acid / chitosan / polyacrylamide, referred to as ZGBCP, for application in aqueous zinc batteries. The authors argue that the electrochemical performance of this electrolyte in a wide temperature range is attributed to the unique solvation properties of ZGBCP: (a) exchanging water from the Zn(II) solvation shell by alcohol- and amino-functionalities of glycerol/chitosan/polyacrylamide, (b) suppressing BF_4^- anion hydrolysis by catching fluoride and protons with boric acid, and (c) providing mechanical stability, adhesiveness and ionic transport with the polymeric network of chitosan and polyacrylamide.

General comments:

It is unclear why the strategy is called co-reconstruction.

All experimental electrolyte systems (ZGP, ZCP and ZGBCP) were tested with a single composition ratio, respectively. It seems the authors have drawn mechanistic conclusions based on this single composition, without showing trends for varying amounts of the different compounds in these multi-component electrolytes.

How do the different hydrogel electrolytes compare in water content expressed

L82: by "protonation", do the authors refer to hydrogen-bonding as a non-covalent crosslinking mechanism?

References for Gaussian, Multiwfn, DFT functionals/basis sets and solvation models are missing.

Critical comments:

Details about the MD simulations, such as structures of all species and their numbers are missing. The validity of the MD simulations is also not addressed. How do modelled transport properties, such as diffusivity, compare to experiments?

Even though MD simulations are conducted that allow to collect ensembles of different complete solvation complexes, DFT energies are only computed for strongly simplified, partial solvation shells, even comparing absolute energies of mono-dentate and bidentate complexes directly.

The DFT functional used is outdated. Given the size of the complexes if sampled from the MD simulation, B97-3c is a newer, much more accurate and cheaper alternative for optimization.

Polyacrylamide was simplified to a trimer, however, its structure doesn't match the polymer backbone. There should be a terminal CH₂, which can be saturated to CH₃ to cap the polymer. Further, no explanation is given as to why a trimer was also used in the classical MD simulations.

It is unclear how chitosan was depicted in the simulations. The possible complexation of ions by chitosan wasn't investigated at all.

In summary, for the analysis of the mechanistic effects in the hydrogel electrolytes, the authors conducted computational studies, which methodology appears to be misleading and/or incomplete. Therefore, the work does not support the conclusion and claims made by the authors.

Response to Reviewers' Comments

Reviewer #1 (Remarks to the Author):

This work utilizes a multi-component electrolyte with $Zn(BF_4)_2$ -glycerol-boric acid-chitosan-polyacrylamide and achieves good performance over a wide temperature range of -50 to 100 °C. A new solvation view is proposed, highlighting reconstructing the anionic and cationic solvation structures, which contribute to high ionic conductivity and cation transfer numbers. The proposed structure is interesting, but comparisons with similar works in gel or liquid electrolytes are lacking. The complexity of the electrolyte composition (with over 3 components except salt) may lead to results or discussions that do not fully explain the real case. Here are some concerns and suggestions for improvement:

Our response: We appreciate the reviewer for recognizing the importance and novelty of our work and constructive revision advice. We have addressed the reviewer's comments point by point.

1. The claim in sentence 80 about the robust interphase of the ZGBCP electrolyte and the dual crosslinking network involving esterification, protonation, and polymerization reactions needs further explanation to clarify how these reactions contribute to the function.

Our response: Thank you very much for your insightful comments and profound questions. First, esterification between glycerol and boric acid can enhance polymer mobility effectively^{1,2} and then promote the infiltration of ZGBCP into the electrode interface considerably; Second, protonation of chitosan can render the ZGBCP superb adhesion capability, thereby generating extremely intimate interfacial contact; Third, polymerization of acrylamide can provide a robust framework for both the esterification and the protonation and increase the crosslinking density, ultimately leading to a multi-component and multi-function crosslinking network, which contributes the stable

interface between the ZGBCP electrolyte and the electrodes. More detailed information is offered to further elaborate the three parts as follows:

The adhesion properties of the PAM/CS dual crosslinking network mainly stem from the interactions between functional groups such as $-\text{CONH}_2$, $-\text{OH}$, and $-\text{NH}_2$ and the material surface, such as hydrogen bonding and coordination, as previously reported³. However, when serving under extreme temperature conditions, further improvement of interfacial contact is required to ensure sufficient infiltration and high reversibility of electrode reactions. We believe that the challenge of further enhancing its adhesion strength lies in the limited dynamic diffusion of these functional groups due to slow chain segment motion, making it difficult to form a dynamically and consistently robust adhesion⁴.

To address this, we introduced the dissociation of protons by the reversible reaction process between boric acid and glycerol to dissolve CS, thus constructing a PAM/CS dual crosslinking network. Rheological test results indicate that the energy storage modulus G' of the double crosslinked network GBCP prepared by plasticization through reversible esterification reaction between boric acid and glycerol is reduced to 4.4 kPa, whereas that of the double crosslinked network CP prepared using traditional acetic acid strategy is 13.7 kPa. This reflects that the GBCP hydrogel is more prone to dissipating energy through deformation flow, thus facilitating the formation of lasting and robust adhesion with the electrode interface. As shown in Fig. 1 and Supplementary Fig. 4-6, this esterification reaction can enhance the adhesion performance of the PAM/CS dual crosslinking network, thereby improving the infiltration of the hydrogel into the electrode.

Fig. R1 | The variation of storage modulus G' and loss modulus G'' with strain oscillation of PAM/CS crosslinking network with/without the boric acid and glycerol.

More importantly, boric acid and glycerol synergistically modulate the solvation structures, suppressing the HER during zinc plating, avoiding the excessive protons introduced by traditional acetic acid methods, and effectively alleviating the vicious hydrolytic corrosion of free BF_4^- anions, thus avoiding the hydrogen evolution problem of traditional gel electrolytes. Detailed descriptions of this anion/cation solvation structure reconstruction are provided in Fig. 2-3 and Supplementary Fig. 10-14.

In Revision: The sentence 150 was changed to “Fundamentally, this can be attributed to the multiple electrostatic and weak intermolecular interactions induced by the esterification and protonation process in the ZGBCP electrolyte (Fig. 1g)⁵⁻⁸. Besides, the faster motion of chain segment tuned by the esterification also helps the adhesive groups (-OH, -NH₂, and -CONH₂) diffuse dynamically to form a consistently robust adhesion. In addition, the polymerization of AM can enhance the crosslinking density, ensuring the reinforced stability between the electrolyte and the electrode. In summary, all these three reactions are all essential for the robust interface of the ZGBCP electrolyte with electrodes.” to clarify the contributions of three kinds of reactions to the robust interphase of the ZGBCP electrolyte and the dual crosslinking network. Besides, Fig. 1b was replaced by the initial Fig. 1c, and Fig. 1c was replaced by Fig. R1 to further clarify the effects induced by esterification and protonation.

2. The reconstruction of cationic and anionic solvation structures is emphasized but not clearly summarized in the paper.

Our response: Thank you for your good advice. First, for the reconstruction of cationic solvation structure, $-\text{OH}$ in glycerol and $-\text{CONH}_2$ in the PAM backbone show stronger affinity to Zn^{2+} than the water molecules, leading to the partial replacement of water molecules in the cation solvation structure and then forming a new cation solvation composition of $\text{Zn}(\text{BF}_4^-)_{1.4}(\text{H}_2\text{O})_{3.6}(\text{C}_3\text{H}_8\text{O}_3)_{0.6}\text{PAM}_{0.4}$ in the ZGBCP electrolyte instead of the $\text{Zn}(\text{BF}_4^-)_{0.8}(\text{H}_2\text{O})_{5.2}$ in traditional AE, which effectively suppresses HER at the Zn electrode and accelerates the stripping/plating kinetics of the Zn electrode. Second, for the reconstruction of anionic solvation structure, similar to that of the above cationic solvation structure, $-\text{OH}$ in glycerol, $-\text{OH}$ in boric acid, and $-\text{CONH}_2$ in the PAM backbone act as hydrogen bond donors, thus replacing water molecules around free BF_4^- anions and forming new solvation structures, such as $[\text{BF}_4^-(\text{H}_2\text{O})(\text{H}_3\text{BO}_3)(\text{C}_3\text{H}_8\text{O}_3)]^-$. Unlike the vicious hydrolysis reaction of BF_4^- induced by the surrounding water molecules in the AE, the BF_4^- can be stabilized by the ZGBCP electrolyte. The reconstructed solvation structures of both the cations and anions together contributes faster electrochemical charge transfer reaction, more excellent rate performance, wider temperature range and longer cycle life compared with the traditional aqueous electrolyte and other reported electrolyte systems.

In Revision: The sentences of 'In conclusion, based on the dual cross-linking framework with robust adhesion and faster motion of chain segments as designed, the reconstruction of the cationic solvation structure enables the ZGBCP electrolyte to suppress the vicious HER and accelerate the stripping/plating kinetics of Zn anodes. Furthermore, the anionic solvation structures are reconstructed to prevent the corrosion induced by the BF_4^- decomposition and accelerate the charge transfer process towards cathodes, enabling the novel ZGBCP electrolytes for ultra-stable and ultrawide-temperature-range AZMBs (Fig. 5g).'' was provided in the discussion part to provide more detailed explanation of reconstruction of solvation structures.

3. The statement about the strong hydrogen bond between central hydroxyl-H and one of the edge hydroxyl-O within the glycerol molecule contributing to rapid desolvation lacks evidence to support it.

Our response: Thank you very much for your profound question. First, in previous studies about polyols, M. Mostafavi et al. spectroscopically confirmed the rapid solvation/desolvation kinetics contributed by the glycerol⁹. Second, to provide further evidence to support our hypothesis, we systematically investigated the roles of similar compounds such as 1,2-propanediol and 1,3-propanediol with different intramolecular hydrogen bonding in regulating the desolvation. As seen from Fig. R2a and R2b, Raman spectroscopy reveals that the intensity of the symmetric and asymmetric stretching vibrations of O–H bonds in glycerol is much higher than those of 1,2-propanediol and 1,3-propanediol, indicating the stronger coupling of O–H vibrational modes caused by more intense intramolecular hydrogen bonding of glycerol¹⁰. Furthermore, to identify the influence of the intramolecular hydrogen bonding on the desolvation process, CA test of the Zn electrode with the electrolytes containing glycerol, 1,2-propanediol and 1,3-propanediol was conducted. It was observed from Fig. R2c that the Zn²⁺ transfer number of ZGP electrolyte with glycerol is 0.61, whereas for electrolytes (Z12PP and Z13PP) using 1,2-propanediol and 1,3-propanediol, the Zn²⁺ transfer number are only 0.25 and 0.33, respectively, proving that glycerol did accelerate the Zn desolvation as well as Zn plating reaction. Due to the accelerated desolvation process, Zn|Zn symmetric cell using ZGP electrolyte exhibits the lowest initial plating polarization voltage at a current density of 1 mA cm⁻², shown in Fig. R2d.

Fig. R2 | Spectral and electrochemical characterization of the hydrogen-bond effects in different polyols. **a-b**, Raman spectra of the pure glycerol, 1,2-dipropanol, and 1,3-dipropanol. **c**, Chronoamperometry tests, and **d**, Voltage-time profiles of Zn|Zn symmetric cells using ZGP, Z12PP, and Z13PP electrolytes containing glycerol, 1,2-dipropanol, and 1,3-dipropanol, respectively.

In Revision: As advised by the reviewer, to prove the contribution of strong hydrogen bonding in the glycerol to the rapid desolvation reaction, Fig. R2b, R2c, and R2d were provided as Supplementary Fig. 19c-e in the supporting file and the explanation ‘To further support the hypothesis, the roles of similar compounds such as 1,2-propanediol and 1,3-propanediol with different intramolecular hydrogen bonding in regulating the desolvation were systematically investigated. As seen from Supplementary Fig. 19c, Raman spectroscopy reveals that the intensity of the symmetric and asymmetric stretching vibrations of O–H bonds in glycerol is much higher than those of 1,2-propanediol and 1,3-propanediol, indicating the stronger coupling of O–H vibrational modes caused by more intense intramolecular hydrogen bonding of glycerol¹⁰. Furthermore, to identify the influence of the intramolecular hydrogen bonding on the desolvation process, CA test of the Zn|Zn cells with the

electrolytes containing glycerol, 1,2-propanediol and 1,3-propanediol were conducted. It was observed from Supplementary Fig. 19d that the Zn^{2+} transfer number of ZGP electrolyte with glycerol is 0.61, whereas for electrolytes (Z12PP and Z13PP) using 1,2-propanediol and 1,3-propanediol, the Zn^{2+} transfer number are only 0.25 and 0.33, respectively, proving that glycerol did accelerate the Zn desolvation as well as Zn plating reaction. Due to the accelerated desolvation process, Zn|Zn symmetric cell using ZGP electrolyte exhibits the lowest initial plating polarization voltage at a current density of 1 mA cm^{-2} , shown in Supplementary Fig. 19e.' was supplemented in the supporting file.

4. The reliability of transfer number measurements is questioned due to potential influences from electrode surfaces and applied voltages. More evidence or alternative methods, such as calculating desolvation energy of cations, should be provided.

Our response: Thank you very much for your insightful comments. Detailed tests were conducted using EIS under different temperatures, as shown in Fig. R3a (namely Fig. 2d in the main text). The activation energy E_a obtained through fitting the Arrhenius relationship includes the energy barrier of the desolvation process. Therefore, this can indirectly reflect the desolvation energy barrier of different electrolytes. As advised by the reviewer, to further prove this conclusion directly, we employed the w97-b3c functional to perform DFT calculations on the main solvation structures of ZGBCP and AE electrolytes obtained from molecular dynamics simulations. As shown in Fig. R3b, the desolvation energy barrier of $[Zn^{2+}BF_4^-(H_2O)_5]^+$ is as high as 8.15 eV, while that of $[Zn^{2+}BF_4^-(H_2O)_4(C_3H_8O_3)]^+$ -PAM is only 3.46 eV, confirming the experimental trend of significantly reduced desolvation energy barriers in the ZGBCP electrolyte.

Fig. R3 | (a) The activation energy barrier based on the fitting of $\ln R_{ct}^{-1}$ versus $1000/T$ Arrhenius curves of different electrolytes. (b) Desolvation energy barrier ΔE_{desol} of $[\text{Zn}^{2+}(\text{BF}_4^-)(\text{H}_2\text{O})_4(\text{C}_3\text{H}_8\text{O}_3)]^+-\text{PAM}$ and $[\text{Zn}^{2+}(\text{BF}_4^-)(\text{H}_2\text{O})_5]^+$ obtained from the DFT calculation.

In Revision: As advised by the reviewer, DFT calculations of the desolvation energy were provided as the direct evidence for the reliability of transfer number result. Fig. R3b was added as Supplementary Fig. 19f to the supporting file. The sentence ‘Besides, the stronger anionic affinity of boric acid and glycerol is also conducive to the transfer process of cations, which interprets the high transfer number (0.82) and much lower desolvation energy barrier (3.46 eV) of Zn^{2+} in the ZGBCP electrolyte as shown in above Fig. 2g and Supplementary Fig. 19f.’ was added for further explanation.

5. The differences in solvation structures of cations and anions compared to other reported electrolytes are not clearly demonstrated in the current simulation results. The reconstruction of anion solvation structures should be more clearly presented.

Our response: Thank you for your suggestion. We have extensively reviewed relevant literature, and a comparison of different Zn^{2+} solvation structures was provided and shown in Table R1. It is clear that the previously developed strategy mainly focused on altering the cation solvation structure by adding high concentration salts or organic solvents, while there were no discussions on the solvation structures of the BF_4^- anions. Fig.S12-13 elaborate on the changes in anion solvation structures and the distribution of the surrounded species in the ZGBCP electrolyte and traditional aqueous electrolyte.

The anionic solvation structure is mainly governed by weak interactions of hydrogen bonding. First, the function groups of $-\text{OH}$ in glycerol, $-\text{OH}$ in boric acid, and $-\text{CONH}_2$ in polyacrylamide acting as hydrogen bond donors replace the water molecules around free BF_4^- . Second, these hydrogen bond donors can also confine the free water molecules and inhibit the water-induced vicious hydrolysis reaction of BF_4^- . Thereby, the reconstruction of anions can be realized. To further answer your question, we analysis all the potential anionic solvation structures as shown in Fig. R4 and R5. It reveals that all the reconstructed structures exhibit lower solvation energy and narrower energy gap, thus the charge transfer process during cycling can be accelerated on the basis of protecting free BF_4^- from the attacking of H_2O .

Fig. R4 | Solvation energy of different solvation structures of uncoordinated anions denoted as $\text{BF}_4^-(\text{H}_2\text{O})_a(\text{C}_3\text{H}_8\text{O}_3)_b(\text{H}_3\text{BO}_3)_c(\text{PAM})_d(\text{CS})_e$.

Fig. R5 | Energy level of different solvation structures of uncoordinated anions denoted as $\text{BF}_4^-(\text{H}_2\text{O})_a(\text{C}_3\text{H}_8\text{O}_3)_b(\text{H}_3\text{BO}_3)_c(\text{PAM})_d(\text{CS})_e$.

Table. R1 | The comparisons of solvation structures and performances of the reported electrolytes based on the $Zn(BF_4)_2$

Ref	gel or liquid	Cationic solvation structure (representative)	Anionic solvation structure (representative)	Strategy	Overpotential under 1 mA cm^{-2} (mV)	Working temperature ($^{\circ}C$)
11	liquid	$[Zn^{2+}(BF_4^-)_2(DME)(H_2O)_2]$	/	Organic Solvent	~170	+25
12	liquid	$[Zn^{2+}(BF_4^-)(H_2O)_2(ace)_3]^+$	/	Organic Solvent	~80	0-+25
13	liquid	$[Zn^{2+}(BF_4^-)_2(H_2O)_4]^+$	/	Organic Solvent	/	-60-+25
14	liquid	$[Zn^{2+}(BF_4^-)_2(H_2O)_2(TMP)_2]^+$	/	Organic Solvent	200	+25
15	Gel	$[Zn^{2+}(BF_4^-)_2(H_2O)_2(PAM)]^+$	/	Saturated salts	/	-70-+25
This work	Gel	$[Zn^{2+}(BF_4^-)(H_2O)_4(C_3H_8O_3)]^+$ - PAM	$[BF_4^-(H_2O)(C_3H_8O_3)(H_3BO_3)]^-$	CRACCS	87	-50-+100

In Revision: As advised by the reviewer, in the revised manuscript, we further discuss the reconstruction of the anionic solvation structure in the main text. The Table R1, Fig. R4 and R5 were added to the supporting file and the following explanation ‘The anionic solvation structure is primarily governed by weak interactions of hydrogen bonding. First, the function groups of -OH in glycerol, -OH in boric acid, and -CONH₂ in polyacrylamide acting as hydrogen bond donors replace the water molecules around free BF_4^- . Second, these hydrogen bond donors can also confine the free water molecules and inhibit the water-induced vicious hydrolysis reaction of BF_4^- . Thereby, the reconstruction of anions can be realized.’ was added into the main text.

Reviewer #2 (Remarks to the Author):

The paper titled "Co-reconstruction of anionic and cationic solvation structures for ultra-stable operating of zinc-metal batteries under -50~+100oC" by authors L. Yao, J. Liu, F. Zhang, B. Wen, X. Chi and Y. Liu deals with the regulation of cationic/anionic solvation structures in aqueous zinc metal batteries by using multicomponent electrolyte, consisted of Zn(BF₄)₂, glycerol, boric acid, chitosan and polyacrylamide (ZGBCP). The authors presented a detailed experimental and computational investigation of solvation structure and a comprehensive understanding of the mechanism in proposed battery devices. Also, the performances of proposed water-based zinc batteries are promising. All the experiments are well-thought-out and well-designed resulting in a coherent story even though the investigated system is complicated. Overall, the article is interesting, actual, and novel.

Our response: We appreciate the reviewer for recognizing the novelty and importance of our work and constructive advice. We have carefully addressed the reviewer's comments point by point.

Recommendation: Major revision

Below are some comments, hope will help to modify this paper:

Major remarks:

1.DFT calculations. The use of DFT calculations to explain the coordination of Zn²⁺ and BF₄⁻ in terms of binding energies and bond order is a great idea.

Is there any particular reason why chitosan is excluded from the discussion and calculation of binding energies? From Scheme 1g, chitosan is proposed to be one of the coordination sites for Zn-ion, along with acrylate, which is a reasonable assumption due to the existence of -OH and -NH groups in the chitosan structure. However, there is no data for Zn²⁺-chitosan binding energies from DFT calculation which is contradictive. Also, does BF₄⁻ have any interactions with chitosan or PAM based on DFT calculations? Please, revise the DFT models and change the proposed solvation model accordingly.

Our response: Firstly, thank you very much for your insightful question. We agreed with the reviewer that chitosan (CS) has interactions with Zn^{2+} . So, the CS was actually taken into account into the calculation system. However, CS constitutes only 4.1 wt% of the entire electrolyte framework as mentioned in the experimental section, thus showing little influence on the average solvation structure from the calculation results. Therefore, the discussions of the DFT calculation sections mainly focused on the interactions of PAM with Zn^{2+} . According to the reviewer's advice, we have modified the calculation model and enhanced the calculations of the interactions between PAM, CS, Zn^{2+} , and BF_4^- . The new calculation results are shown in Fig R6 and Fig R7. Compared to PAM, chitosan exhibits slightly stronger affinity for Zn^{2+} (-1.57 eV vs. -1.24 eV). Functional groups $-\text{OH}$ and $-\text{NH}$ in CS have higher degrees of freedom than amide group in PAM, and the N atom in the $-\text{NH}_2$ group possesses lone pair electrons, enabling it to act as a Lewis base and form coordination bonds with metal ions more effectively¹⁶. As to the anion interactions, the calculation results in Fig. R7 show that CS exhibits a stronger affinity to BF_4^- (-0.28 eV for PAM vs. -0.41 eV for CS). This interaction is dominated mainly by hydrogen bonding of BF_4^- with hydrogen bonding donors such as $-\text{NH}_2$, and $-\text{OH}$ in the CS.

Fig. R6 | Binding energies between Zn^{2+} with PAM and CS.

Fig. R7 | Binding energies between BF₄⁻ with PAM and CS.

In Revision: As advised by the reviewer, in the revised supporting file, we supplemented the Fig. R6 and Fig. R7; in the revised manuscript, we provided more discussions of the interactions between the CS and cations/anions.

2.MD simulations. It is difficult to analyze presented RDFs data from MD simulations and to evaluate the validity of the used force field. Authors should provide the number of molecules (in Supporting information) and the size of the simulation box used in MD simulations during the production run. Did the authors perform any validation of the used force field? I will strongly recommend showing the obtained data in the form of coordination number since presented RDFs (Figure 2e) are impossible to analyze and therefore the corresponding discussion is confusing. Also, please note that chitosan is again missing like in the DFT calculations, so the proposed discussion is ambiguous and needs to be revised accordingly.

Our response: Thank you for your profound comments and questions. For molecular dynamics (MD) simulations, the size of the simulation box and numbers of molecules are presented in the following Table R2.

Table R2. | Box length and molecule numbers of ZGBCP and AE electrolyte systems.

System	Box	H ₂ O	Zn(BF ₄) ₂	C ₃ H ₈ O ₃	H ₃ BO ₃	PAM	CS
	length (Å)						
ZGBCP	52	1520	118	128	57	55	1
AE	49	2861	206	\	\	\	\

As to the force field, GAFF force field was applied in this work since it has been widely used for molecular dynamics simulations of electrolytes^{11, 17, 18}. In the ZGBCP system, the optimal solvation structure of Zn^{2+} obtained from MD simulations is $\text{Zn}(\text{BF}_4^-)_{1.4}(\text{H}_2\text{O})_{3.6}(\text{C}_3\text{H}_8\text{O}_3)_{0.6}\text{PAM}_{0.4}$, while in the AE system, it is $\text{Zn}(\text{BF}_4^-)_{0.8}(\text{H}_2\text{O})_{5.2}$. To validate its effectiveness, DFT calculations were systematically performed for various potential solvation structures as shown in Fig R8. The solvation structure under primary discussion exhibited the lowest energy, thereby confirming the validity of the MD simulation. Additionally, MD simulations were conducted at different temperatures, and the simulated conductivities based on Mean Square Displacement (MSD) show consistent trends and magnitudes with experimental results, further confirming the validity of the simulation as seen in Fig. R9.

Fig. R8 | Structures of potential solvation structures of Zn^{2+} and corresponding solvation energy E_{sol} in the (a) AE, and (b) ZGBCP electrolyte.

Fig. R9 | Ionic transport property calculated by the MD simulation under different temperatures. **a**, MSD of Zn^{2+} obtained from MD simulations under different temperatures. **b**, Diffusion coefficients and **c**, Conductivities of ZGBCP electrolytes based on simulation and experiments under different temperatures.

For the question of coordination configuration and the influence of CS, as mentioned above, CS constitutes only 4.1wt% of the whole framework, thus exerting little influence on the average solvation structure as seen in the Fig. R10a. Further, as shown in Supplementary Fig. 2, unlike other zinc salts such as ZnCl_2 and $\text{Zn}(\text{ClO}_4)_2$, the typical CIP structure of $\text{Zn}(\text{BF}_4)_2$ electrolyte makes it difficult to induce CS gelation through divalent ion crosslinking. To further address your inquiry, we investigated the solvation structure near CS in the ZGBCP system. The results as shown in Fig. R10 indicate the formation of a solvation structure: $[\text{Zn}(\text{BF}_4^-)_2(\text{H}_2\text{O})_2(\text{C}_3\text{H}_8\text{O}_3)]\text{-CS}$. Detailed DFT calculations were then conducted about this structure and the ESP results show that the dissociation energy barrier of H in coordinating water molecules increase over 2 eV, corresponding to a more reduced tendency for HER (Fig. R11 and R12).

Fig. R10 | (a) $g(r)$ and $n(r)$ of overall Zn^{2+} from CS. (b) $n(r)$ of the Zn^{2+} coordinated by CS.

Fig. R11 | The ESP distribution of $[\text{Zn}^{2+}(\text{BF}_4^-)_2(\text{H}_2\text{O})_2(\text{C}_3\text{H}_8\text{O}_3)]\text{-CS}$ in the ZGBCP electrolyte and $[\text{Zn}^{2+}(\text{BF}_4^-)(\text{H}_2\text{O})_5]^+$ in the AE electrolyte.

Fig. R12 | Electrostatic interaction energy of the H sites in H_2O of $[\text{Zn}^{2+}(\text{BF}_4^-)_2(\text{H}_2\text{O})_2(\text{C}_3\text{H}_8\text{O}_3)]\text{-CS}$ in the ZGBCP electrolyte and $[\text{Zn}^{2+}(\text{BF}_4^-)(\text{H}_2\text{O})_5]^+$ in the AE electrolyte.

In Revision: As advised by the reviewer, in the revised supporting file, we supplemented the Table. R2 and Fig. R8-R12. in the revised manuscript, we have incorporated more discussions on the coordination number and clarified the interactions within the cationic solvation structures and roles of the chitosan in the electrolyte.

3. Please, provide the procedure for treatment of the electrode before post-mortem analysis. SEI formation could be easily influenced by post-treatment of the electrode, and it is necessary to include a detailed handling procedure of the electrode.

Our response: We agree with the reviewer's comment on the treatment of the electrode. The treatment procedure is as followings: first, the gel residues adhering to the surface of the cycled zinc foil was completely peeled off with great care. Given the good solubility of $\text{Zn}(\text{BF}_4)_2$ in ethanol and the relatively low boiling point of ethanol, the zinc

foil cycled with ZGBCP and AE electrolytes were then rinsed with anhydrous ethanol to prevent any potential influence of secondary hydrolysis of BF_4^- on the determination of F species in the SEI. Subsequently, the samples were vacuum dried, transferred under inert atmosphere protection, and promptly subjected to relevant tests to avoid secondary oxidation and radiation damage.

In Revision: The treatment procedure ‘First, the gel residues adhering to the surface of the cycled zinc foil was completely peeled off with great care. Given the good solubility of $\text{Zn}(\text{BF}_4)_2$ in ethanol and the relatively low boiling point of ethanol, the zinc foil cycled with ZGBCP and AE electrolytes were then rinsed with anhydrous ethanol to prevent any potential influence of secondary hydrolysis of BF_4^- on the determination of F species in the SEI. Subsequently, the samples were vacuum dried, transferred under inert atmosphere protection, and promptly subjected to relevant tests to avoid secondary oxidation and radiation damage.’ was added to the experimental part.

Minor remarks:

4. XPS data. It would be interesting to include a wide range of data for XPS measurements in Supporting information. The shift in XPS data is obvious (Figure 3e and Fig. S14). Did the authors perform any internal calibration of XPS data?

Our response: We sincerely appreciate your good comments. As advised, we have provided detailed XPS full spectra and C1s peak data corrected for charging effects at different depths (Fig. R13). Besides, the shift in XPS data was also calibrated and the revised figures were shown in Fig. R14 and R15. Your feedback is crucial for ensuring the accuracy of our data, and we are grateful once again for your meticulous inquiries.

Fig. R13 | The XPS survey and corresponding C 1s spectra of Zn anodes at different depths using (a) ZGBCP, and (b) AE electrolyte.

Fig. R14 | XPS spectra of Zn 2p of Zn anodes at different depths using (a) ZGBCP, and (b) AE electrolyte.

Fig. R15 | (a) XPS survey and (b) F 1s spectra of ZGBCP, ZGP, and ZCP electrolytes.

In Revision: The Fig. R13 was added to the revised supporting file. The original Fig. 2f was replaced with the Fig. R14 and Fig. R15 was provided as Fig. 22 in the revised supporting file, respectively.

5. English should be improved in the entire Manuscript.

Our response: We sincerely appreciate the valuable feedback from the reviewer. We have meticulously reviewed the entire article multiple times and corrected the language errors and inaccuracies to ensure clarity and precision in its language expression. The revised parts have been highlighted in both the main text and the supporting file. We are grateful for the reviewer's suggestions and are committed to putting in the effort to ensure that every aspect of the manuscript reaches the highest standards.

6. Chapter before "Methods" should be Conclusion, not Discussion.

Our response: Thank you for your suggestion. The Conclusion was actually included in the Discussion part since the *Nature Communications* journal advises to provide a

comprehensive Discussion part. Combined with the reviewer's advice, the Chapter before 'Method' was changed to 'Conclusion and Discussion'. We appreciate the rationale behind it and believe that this adjustment will indeed better organize the structure of the article. We are truly grateful for the insightful guidance you have provided, and we are confident that this adjustment will enhance the coherence and readability of the article. Once again, thank you for your invaluable input.

Reviewer #3 (Remarks to the Author):

The manuscript by Yao et al introduces a novel quinary electrolyte of Zn(BF₄)₂/ glycerol / boric acid / chitosan / polyacrylamide, referred to as ZGBCP, for application in aqueous zinc batteries. The authors argue that the electrochemical performance of this electrolyte in a wide temperature range is attributed to the unique solvation properties of ZGBCP: (a) exchanging water from the Zn(II) solvation shell by alcohol- and amino-functionalities of glycerol/chitosan/polyacrylamide, (b) suppressing BF₄⁻-anion hydrolysis by catching fluoride and protons with boric acid, and (c) providing mechanical stability, adhesiveness and ionic transport with the polymeric network of chitosan and polyacrylamide.

Our response: We appreciate the reviewer for recognizing the innovations of our work and proposing very constructive revision advice. We have carefully and thoroughly addressed the reviewer's comments below.

General comments:

1.It is unclear why the strategy is called co-reconstruction.

Our response: Thank you so much for your insightful questions. We sincerely apologize for the unclear definition. The co-reconstruction represents the simultaneous reconstruction of cationic (Zn²⁺) and anionic (BF₄⁻) solvation structures, simplified as CRACSS. To better and more clearly explain the unique solvation structure modification strategy, we have added the following explanations in the revised manuscript. First, for the reconstruction of cationic solvation structure, –OH in glycerol and –CONH₂ in the polyacrylamide backbone shows strong affinity to Zn²⁺, leading to the partial replacement of water molecules in the cationic solvation structure and then forming a new cation solvation composition of Zn(BF₄⁻)_{1.4}(H₂O)_{3.6}(C₃H₈O₃)_{0.6}PAM_{0.4} in the ZGBCP electrolyte instead of the Zn(BF₄⁻)_{0.8}(H₂O)_{5.2} in traditional aqueous electrolyte, which effectively suppresses HER at the Zn electrode and accelerates the stripping/plating kinetics of the Zn electrode. Second, for the reconstruction of anionic solvation structure, similar to that of the above cationic solvation structure, –OH in

glycerol, $-OH$ in boric acid, and $-CONH_2$ in the polyacrylamide backbone act as hydrogen bond donors, thus replacing water molecules around free BF_4^- anions. These hydrogen bond donors can also confine the free water molecules and inhibit the water-induced vicious hydrolysis reaction of BF_4^- and stabilize the BF_4^- by the ZGBCP electrolyte. The reconstructed solvation structures of both the cations and anions together contributes faster electrochemical charge transfer reaction, more excellent rate performance, wider temperature range and longer cycle life compared with the traditional aqueous electrolyte and other reported electrolyte systems.

In Revision: The above explanations were added to the Discussion part in the revised manuscript.

2.All experimental electrolyte systems (ZGP, ZCP and ZGBCP) were tested with a single composition ratio, respectively. It seems the authors have drawn mechanistic conclusions based on this single composition, without showing trends for varying amounts of the different compounds in these multi-component electrolytes.

Our response: Thank you for your insightful question. To save space in the article, the description of the gradient orthogonal experiment was omitted. Firstly, the dosage of PAM and CS was referenced from the typical formulations of double cross-linking network electrolyte¹⁹. Besides, the feeding amount of boric acid (BA, 0.6 g) nearly reached its solubility limit, while an appropriate amount of glycerol ensured the occurrence of esterification reaction, achieving a balance between adhesiveness, conductivity, and mechanical properties. Table. R3 provides the exchange current density of zinc stripping/plating reactions of electrolytes with different amounts of boric acid and glycerol, demonstrating that the composition of ZGBCP electrolyte exhibits the fastest zinc stripping/plating kinetics.

Table. R3 | The exchange current densities (mA/cm^2) of Zn stripping/plating reactions in different electrolytes containing varied amounts of boric acid (BA) and glycerol.

Boric acid (g) \ Glycerol (g)	0.2	0.4	0.6	0.8
2	0.049	0.040	0.057	BA precipitation
4	0.045	0.043	0.032	BA precipitation
6	CS insoluble	CS insoluble	CS insoluble	BA precipitation

In Revision: The Table R3 was provided in the revised supporting file and the corresponding explanation for the optimization of the electrolyte composition was added in the revised manuscript.

3.How do the different hydrogel electrolytes compare in water content expressed.

Our response: Thank you for your good question. As advised, thermal gravimetric analysis (TGA) was conducted to determine the water content of different hydrogels. The results were compared in the Fig. R16. It can be seen that the water contents of the ZCP, ZGP and ZGBCP hydrogel electrolytes were 34.2 wt%, 51.6 wt%, and 28.1 wt%, respectively.

Fig. R16 | TGA curves of ZGBCP, ZCP, and ZGP electrolytes after soaking in 2 M $\text{Zn}(\text{BF}_4)_2$ electrolytes.

In Revision: The Fig. R16 was supplemented in the revised supporting file.

4.L82: by “protonation”, do the authors refer to hydrogen-bonding as a non-covalent crosslinking mechanism?

Our response: We are very sorry for the unclear explanation. The chitosan (CS) generally requires acid, e.g. the commonly used acetic acid, to dissolve and form a gel in the water. The dissolution and gelation process of CS is a proton-involved process. Herein, we did not add the acid during the electrolyte synthesis. The protons come from the esterification reaction between boric acid and glycerol according to the literature²⁰⁻²², and the motion of chain segments is accelerated through the reaction compared with traditional methods. As shown in Fig. R1, rheological test results indicate that the energy storage modulus G' of the double crosslinked network GBCP prepared by

plasticization through reversible esterification reaction between boric acid and glycerol is reduced to 4.4 kPa, whereas that of the double crosslinked network CP prepared using traditional acetic acid strategy is 13.7 kPa. This reflects that the GBCP hydrogel exhibits more viscoelastic properties, which is more prone to dissipating energy through deformation flow, and the faster motion of chain segments ensures the faster ions transport and helps the adhesive groups diffuse dynamically to form a consistently robust adhesion. Therefore, the protonation refers to the proton-stimulated dissolution and gelation of CS.

Fig. R1 | The variation of storage modulus G' and loss modulus G'' with strain oscillation of PAM/CS crosslinking network with/without the boric acid and glycerol.

In Revision: The Fig. R1 was provided to clarify the effects induced by esterification and protonation process. Besides, the sentence 150 was changed to “Fundamentally, this can be attributed to the multiple electrostatic and weak intermolecular interactions induced by the esterification and protonation process in the ZGBCP electrolyte (Fig. 1g)⁵⁻⁸. Besides, the faster motion of chain segment tuned by the esterification also helps the adhesive groups (-OH, -NH₂, and -CONH₂) diffuse dynamically to form a consistently robust adhesion. In addition, the polymerization of AM can enhance the crosslinking density, ensuring the reinforced stability between the electrolyte and the electrode. In summary, all these three reactions are all essential for the robust interface of the ZGBCP electrolyte with electrodes.” to clarify the contributions of three kinds of reactions to the robust interphase of the ZGBCP electrolyte and the dual crosslinking

network.

5. References for Gaussian, Multiwfn, DFT functionals/basis sets and solvation models are missing.

Our response: Thank you for the valuable suggestion. We have included the important references (Ref. 65, 69-75) related to the computational tools and theoretical frameworks we used in the revised manuscript.

In Revision: In the revised manuscript, citations (Ref. 65, 69-75) of Gaussian 09 software package, ORCA software package, as well as the Multiwfn program are cited. Additionally, the citations for the detailed information on the selection of the density functional theory (DFT), basis sets, and implicit solvent models used in our DFT calculations were also supplemented. Thank you for your thorough review and sincere suggestions.

6. Details about the MD simulations, such as structures of all species and their numbers are missing. The validity of the MD simulations is also not addressed. How do modelled transport properties, such as diffusivity, compare to experiments?

Our response: Thank you for your constructive advice. Table R2 and Fig. R17 provide detailed information on the box size, species structure, and quantity obtained from the MD simulations. To validate its effectiveness, DFT calculations were systematically performed for various potential solvation structures as shown in Fig R8. The solvation structure under primary discussion exhibited the lowest energy, thereby confirming the validity of the MD simulation. Additionally, we conducted further MD simulations of the ZGBCP system at different temperatures. Based on MSD calculations, diffusion coefficients σ at different temperatures were obtained. The diffusion coefficients obtained from the simulations exhibit similar trends and orders of magnitudes as those obtained from the experimentally measured conductivity versus temperatures as shown in Fig. R18. This confirms the accuracy and reliability of the simulation results.

Table. R2 | Box length and molecule numbers of ZGBCP and AE electrolyte systems.

System	Box	H ₂ O	Zn(BF ₄) ₂	C ₃ H ₈ O ₃	H ₃ BO ₃	PAM	CS
	length (Å)						
ZGBCP	52	1520	118	128	57	55	1
AE	49	2861	206	\	\	\	\

Fig. R17 | Molecular models used in the MD simulations.

Fig. R8 | Structures of potential solvation structures of Zn²⁺ and corresponding

solvation energy E_{sol} in the (a) AE, and (b) ZGBCP electrolyte.

Fig. R18 | Ionic transport property calculated by the MD simulation under different temperatures. **a**, MSD of Zn^{2+} obtained from MD simulations under different temperatures. **b**, diffusion coefficients and **c**, conductivities of ZGBCP electrolytes based on simulation and experiments under different temperatures.

In Revision: In the revised supporting file, the Table R2, Fig. R8, R17 and Fig. R18 and the corresponding explanation were provided. The following explanation ‘To investigate the diffusivity of the ZGBCP electrolyte, MD simulations of the ZGBCP system at different temperatures were further conducted. Based on the MSD calculations (Supplementary Fig. 18), the theoretical diffusion coefficients σ at different temperatures can be obtained. As seen from Supplementary Fig. 18b, the diffusion coefficients and conductivities obtained from the simulations exhibit similar trends and orders of magnitudes as those obtained from the experimentally measured conductivity versus temperatures as shown in Supplementary Fig. 18c This confirms the accuracy and reliability of the simulation results.’ was added in the theoretical calculation part of the revised supporting file.

7. Even though MD simulations are conducted that allow to collect ensembles of different complete solvation complexes, DFT energies are only computed for strongly simplified, partial solvation shells, even comparing absolute energies of mono-dentate and bidentate complexes directly.

Our response: Thank you very much for your insightful perspective and profound questions. MD simulations have been widely recognized to be a reliable tool to gain the insights and a comprehensive view of solvation effects, although some models are simplified. To get much deeper understandings, more extensive computational

resources are needed. Also, the ZGBCP electrolyte developed in this work is a new and relatively complicated system. Due to the limited resources and the uniqueness of the electrolyte, we have tried our best to provide as many calculations as possible in this work. Furthermore, we believe some simplifications used in this work are based on an understanding of the crucial interactions and energy contributions within the new electrolyte system.

To provide more theoretical calculation models and gain a deeper understanding of the new solvation structures built by the CRACSS strategy, first, based on the reviewer's suggestion, we calculated the binding energy of Zn^{2+} and BF_4^- with two H_2O molecules as shown in Fig. R19. Besides, we calculated more cationic solvation structures that are possible in the ZGBCP electrolyte system, which included the CIP-type structures without polymer involvement: $[\text{Zn}^{2+}(\text{BF}_4^-)(\text{H}_2\text{O})_4(\text{C}_3\text{H}_8\text{O}_3)]^+$ and SSIP-type structures: $[\text{Zn}^{2+}(\text{H}_2\text{O})_5(\text{C}_3\text{H}_8\text{O}_3)]^{2+}$ and $[\text{Zn}^{2+}(\text{H}_2\text{O})_6]^{2+}$ based on the B97-3c functional. The solvation structure $[\text{Zn}^{2+}(\text{BF}_4^-)(\text{H}_2\text{O})_4(\text{C}_3\text{H}_8\text{O}_3)]^+$ -PAM under primary discussion exhibited the lowest energy, thereby confirming the validity of the MD simulation as illustrated in Fig. R8. What's more, the additional two solvation structures in the ZGBCP electrolyte also exhibit the inhibited HER tendency and more active electron states (Fig. R20 and R21), which can further suppress the side reactions in the Zn anodes and accelerate charge transfer kinetics as well as the solvation structure under primary discussion.

Fig. R19 | Binding energy of Zn^{2+} and BF_4^- with two H_2O molecules.

Fig. R8 | Structures of potential solvation structures of Zn^{2+} and corresponding solvation energy E_{sol} in the (a) AE, and (b) ZGBCP electrolyte.

Fig. R20 | (a) ESP distribution, (b) energy level, and (c) electrostatic interaction energy (-H) of $[\text{Zn}^{2+}(\text{BF}_4^-)(\text{H}_2\text{O})_4(\text{C}_3\text{H}_8\text{O}_3)]^+$ in the ZGBCP electrolyte and

$[\text{Zn}^{2+}(\text{BF}_4^-)(\text{H}_2\text{O})_5]^+$ in the AE electrolyte.

Fig. R21 | (a) ESP distribution, (b) energy level, and (c) electrostatic interaction energy (-H) of $[\text{Zn}^{2+}(\text{H}_2\text{O})_5(\text{C}_3\text{H}_8\text{O}_3)]^{2+}$ in the ZGBCP electrolyte and $[\text{Zn}^{2+}(\text{H}_2\text{O})_6]^{2+}$ in the AE electrolyte.

In addition, calculations were also performed for various possible solvation structures of BF_4^- anions named as $(\text{H}_2\text{O})_a(\text{C}_3\text{H}_8\text{O}_3)_b(\text{H}_3\text{BO}_3)_c(\text{PAM})_d(\text{CS})_e$. It can be observed from Fig. R4 and R5 that all possible solvation structures of free anions in ZGBCP exhibit lower solvation energy especially the BF_4^- anions near the CS chain, which can effectively protect the free BF_4^- from the attacking of H_2O . What's more all the potential anionic solvation structures exhibit the lower energy gap than the $[\text{BF}_4^-(\text{H}_2\text{O})_3]^-$ in the AE electrolyte, accelerating the charge transfer kinetics with PANI.

Fig. R4 | Solvation energy of different solvation structures of uncoordinated anions denoted as $\text{BF}_4^-(\text{H}_2\text{O})_a(\text{C}_3\text{H}_8\text{O}_3)_b(\text{H}_3\text{BO}_3)_c(\text{PAM})_d(\text{CS})_e$.

Fig. R5 | Energy level of different solvation structures of uncoordinated anions denoted as $\text{BF}_4^-(\text{H}_2\text{O})_a(\text{C}_3\text{H}_8\text{O}_3)_b(\text{H}_3\text{BO}_3)_c(\text{PAM})_d(\text{CS})_e$.

In Revision: In the revised supporting file, the Fig. R4, R5, R8, and R19-21 and the corresponding explanation were provided.

8. The DFT functional used is outdated. Given the size of the complexes if sampled from the MD simulation, B97-3c is a newer, much more accurate and cheaper alternative for optimization.

Our response: Thank you for your sincere suggestions. We have actively adopted your suggestions to improve the efficiency and accuracy of our DFT calculations. By using the B97-3c functional instead of B3LYP, we have recalculated all solvation structures mentioned in the manuscript. We greatly appreciate the valuable feedback from the reviewer, which is crucial for the improvement of our future work.

9. Polyacrylamide was simplified to a trimer; however, its structure doesn't match the polymer backbone. There should be a terminal CH₂, which can be saturated to CH₃ to cap the polymer. Further, no explanation is given as to why a trimer was also used in the classical MD simulations.

Our response: Thank you very much for pointing out the modeling issues. To address this issue, we have re-modeled the PAM to more accurately reflect the polymer's structure. We greatly appreciate the valuable feedback from the reviewer, which is crucial for the improvement of our future work. The new calculation result is shown in Fig. R22.

Fig. R22 | (a) Corrected trimer of PAM. (b) $g(r)$ and $n(r)$ of Zn^{2+} obtained from MD simulation based on the corrected trimer of PAM. (c) ESP of the cationic solvation structures in the ZGBCP and AE electrolytes.

In addition, trimer and pentamer have been widely used in classical molecular dynamics simulations of hydrogel electrolytes^{15, 23-25}. Therefore, we have also conducted molecular dynamics simulations based on pentamer PAM and CS. The simulation results were shown in Fig. R23, Fig. R24 and Table R4, which are similar to those of the trimer MD simulation.

Fig. R23 | Molecular models of PAM and CS pentamers used in the MD simulations.

Table. R4 Box length and molecule numbers of ZGBCP_5mer system.

System	Box length (Angstrom)	H ₂ O	Zn(BF ₄) ₂	C ₃ H ₈ O ₃	H ₃ BO ₃	PAM	CS
ZGBCP_3mer	52	1520	118	128	57	55	1
ZGBCP_5mer	59	2550	197	213	95	55	1
AE	49	2861	206	\	\	\	\

Fig. R24 | (a) $g(r)$ and $n(r)$ of Zn²⁺. (b) $g(r)$ of F in the ZGBCP_5mer system. (c) $n(r)$ of F-H (H₂O) in the ZGBCP_3mer and ZGBCP_5mer systems.

In Revision: In the revised supporting file, the Fig. R23, R24 and Table. R4 as well as the corresponding explanation of ‘As illustrated in Supplementary Fig. 14, the results of MD simulations based on the PAM and CS pentamer models are consistent with those based on the trimer models, verifying the validity of MD simulations.’ were provided.

10.It is unclear how chitosan was depicted in the simulations. The possible complexation of ions by chitosan wasn't investigated at all.

Our response: We apologize for the missing discussions on the simulations and complexation of chitosan. We agreed with the reviewer that chitosan (CS) has interactions with Zn²⁺. So, the CS was actually taken into account into the calculation system. However, CS constitutes only 4.1wt% of the entire electrolyte framework as mentioned in the experimental section, thus showing little influence on the average solvation structure from the calculation results. Therefore, the discussions of the DFT calculation sections mainly focused on the interactions of PAM with Zn²⁺. According to the reviewer’s advice, we have modified the calculation model and enhanced the calculations of the interactions between PAM, CS, Zn²⁺, and especially BF₄⁻. The new

calculation results are shown in Fig. R6 and Fig. R7. Compared to PAM, chitosan exhibits slightly stronger affinity for Zn^{2+} (-1.57 eV vs. -1.24 eV). The functional groups $-\text{OH}$ and $-\text{NH}$ in CS have higher degrees of freedom, and the N atom in the amino group possesses lone pair electrons, enabling it to act as a Lewis base and form coordination bonds with metal ions more effectively¹⁶. As to the anion interactions, the calculation results in Fig. R7 show that CS exhibits a stronger affinity to BF_4^- (-0.28 eV for PAM vs. -0.41 eV for CS). This interaction is dominated mainly by hydrogen bonding of BF_4^- with hydrogen bonding donors such as $-\text{NH}_2$, and $-\text{OH}$ in the CS. In addition, based on the MD simulations, a new solvation structure $[\text{Zn}(\text{BF}_4^-)_2(\text{H}_2\text{O})_2(\text{C}_3\text{H}_8\text{O}_3)]^+-\text{CS}$ is observed near CS chain segment. Furthermore, the DFT calculations about the cationic solvation structures $[\text{Zn}(\text{BF}_4^-)_2(\text{H}_2\text{O})_2(\text{C}_3\text{H}_8\text{O}_3)]^+-\text{CS}$ and potential anionic solvation structures were also conducted. As seen from R10 and R12, the results indicate a significantly increased HER energy barrier and accelerated charge transfer near chitosan coordination (Fig. R25). In addition, calculations were also conducted for various possible solvation structures of BF_4^- anions denoted as $(\text{H}_2\text{O})_a(\text{C}_3\text{H}_8\text{O}_3)_b(\text{H}_3\text{BO}_3)_c(\text{PAM})_d(\text{CS})_e$. It can be observed from Fig. R4 and R5 that all possible solvation structures of free anions in ZGBCP exhibit lower solvation energy especially the BF_4^- anions near the CS chain, which can effectively protect the free BF_4^- from the attacking of H_2O . What's more, all the potential anionic solvation structures exhibit the lower energy gap and higher homo level than the $[\text{BF}_4^-(\text{H}_2\text{O})_3]^-$ in the AE electrolyte, which can accelerate the charge transfer kinetics with PANI cathode.

Fig. R6 | Binding energies between Zn^{2+} with PAM and CS.

Fig. R7 | Binding energies between BF_4^- with PAM and CS.

Fig. R10 | (a) $g(r)$ and $n(r)$ of overall Zn^{2+} from CS. (b) $n(r)$ of the Zn^{2+} near CS.

Fig. R12 | Electrostatic interaction energy ($-H$) of $[\text{Zn}^{2+}(\text{BF}_4)_2(\text{H}_2\text{O})_2(\text{C}_3\text{H}_8\text{O}_3)]-\text{CS}$ in the ZGBCP electrolyte and $[\text{Zn}^{2+}(\text{BF}_4)(\text{H}_2\text{O})_5]^+$ in the AE electrolyte.

Fig. R25 | Energy level of $[\text{Zn}^{2+}(\text{BF}_4)_2(\text{H}_2\text{O})_2(\text{C}_3\text{H}_8\text{O}_3)]\text{-CS}$ in the ZGBCP electrolyte and $[\text{Zn}^{2+}(\text{BF}_4)(\text{H}_2\text{O})_5]^+$ in the AE electrolyte.

Fig. R4 | Solvation energy of different solvation structures of uncoordinated anions denoted as $\text{BF}_4^-(\text{H}_2\text{O})_a(\text{C}_3\text{H}_8\text{O}_3)_b(\text{H}_3\text{BO}_3)_c(\text{PAM})_d(\text{CS})_e$.

Fig. R5 | Energy level of different solvation structures of uncoordinated anions denoted as $\text{BF}_4^-(\text{H}_2\text{O})_a(\text{C}_3\text{H}_8\text{O}_3)_b(\text{H}_3\text{BO}_3)_c(\text{PAM})_d(\text{CS})_e$.

In Revision: As advised by the reviewer, in the revised supporting file, we supplemented the Fig. R4-R7, R10, R12, R25 and the corresponding explanation ‘Although it’s a small amount for CS, a new AGG solvation structure $[\text{Zn}^{2+}(\text{BF}_4^-)_2(\text{H}_2\text{O})_2(\text{C}_3\text{H}_8\text{O}_3)]\text{-CS}$ was found near the CS chain segment, which is conducive to the desolvation process with highest H^+ dissociation energy barrier (Supplementary Fig. 13).’

11. In summary, for the analysis of the mechanistic effects in the hydrogel electrolytes, the authors conducted computational studies, which methodology appears to be misleading and/or incomplete. Therefore, the work does not support the conclusion and claims made by the authors.

Our response:

Thank you for your valuable feedback. We are greatly grateful for your thoughtful consideration of our work. To comprehensively investigate and analyze the completely new electrolyte system, we have tried our best to apply as many characterization tools as possible experimentally and conducted as many modeling methods as possible theoretically. The experimental data and theoretical calculation results show good match and the calculation helps understand the underlying mechanism of the unique properties and performance of the electrolyte. Furthermore, from the theoretical calculation, for the first time, we found a co-reconstruction of both anionic and cationic solvation structures; from the experimental test, for the first time, we demonstrated both an electrolyte with the state-of-the-art working temperature window and a full cell with the best cycling stability. There might be some deficiency in the previous version of the manuscript, however, we believe the revised version has well addressed the reviewer’s concerns and meets the requirements of the journal. Once again, we appreciate your feedback and constructive criticism, which will undoubtedly contribute to enhancing the quality of our manuscript.

Reference

1. Chen M, *et al.* Hydrogen bonding impact on chitosan plasticization. *Carbohydrate Polymers* **200**, 115-121 (2018).
2. Chen P, Xie F, Tang F, McNally T. Unexpected Plasticization Effects on the Structure and Properties of Polyelectrolyte Complexed Chitosan/Alginate Materials. *ACS Applied Polymer Materials* **2**, 2957-2966 (2020).
3. Cintron-Cruz JA, Freedman BR, Lee M, Johnson C, Ijaz H, Mooney DJ. Rapid Ultratough Topological Tissue Adhesives. *Advanced Materials* **34**, 2205567 (2022).
4. Tian G, *et al.* A Nonswelling Hydrogel with Regenerable High Wet Tissue Adhesion for Bioelectronics. *Advanced Materials* **35**, 2212302 (2023).
5. Hong SH, *et al.* Dynamic Bonds between Boronic Acid and Alginate: Hydrogels with Stretchable, Self-Healing, Stimuli-Responsive, Remoldable, and Adhesive Properties. *Biomacromolecules* **19**, 2053-2061 (2018).
6. Fu Q, Hao S, Meng L, Xu F, Yang J. Engineering Self-Adhesive Polyzwitterionic Hydrogel Electrolytes for Flexible Zinc-Ion Hybrid Capacitors with Superior Low-Temperature Adaptability. *ACS Nano* **15**, 18469-18482 (2021).
7. Yu C, *et al.* Chronological adhesive cardiac patch for synchronous mechanophysiological monitoring and electrocoupling therapy. *Nature Communications* **14**, 6226 (2023).
8. Quijada-Garrido I, Iglesias-González V, Mazón-Arechederra JM, Barrales-Rienda JM. The role played by the interactions of small molecules with chitosan and their transition temperatures. Glass-forming liquids: 1,2,3-Propantriol (glycerol). *Carbohydrate Polymers* **68**, 173-186 (2007).
9. Lampre I, Pernet P, Bonin J, Mostafavi M. Comparison of solvation dynamics of electrons in four polyols. *Radiation Physics and Chemistry* **77**, 1183-1189 (2008).
10. Mendelovici E, Frost RL, Klopogge T. Cryogenic Raman spectroscopy of glycerol. *Journal of Raman Spectroscopy* **31**, 1121-1126 (2000).
11. Meng C, He W-D, Tan H, Wu X-L, Liu H, Wang J-J. A eutectic electrolyte for an ultralong-lived Zn//V₂O₅ cell: an in situ generated gradient solid-electrolyte interphase. *Energy & Environmental Science* **16**, 3587-3599 (2023).
12. Wang G, *et al.* Gradient-Structured and Robust Solid Electrolyte Interphase In Situ Formed by Hydrated Eutectic Electrolytes for High-Performance Zinc Metal Batteries. *Advanced Energy Materials* **14**, 2303549 (2024).
13. Wang DD, Peng HL, Zhang SJ, Liu HX, Wang NA, Yang J. Localized Anion-Cation Aggregated Aqueous Electrolytes with Accelerated Kinetics for Low-Temperature Zinc Metal Batteries. *Angewandte Chemie-International Edition*, (2023).
14. Ma G, *et al.* Zn metal anodes stabilized by an intrinsically safe, dilute, and hydrous organic electrolyte. *Energy Storage Materials* **54**, 276-283 (2023).

15. Shi Y, Wang R, Bi S, Yang M, Liu L, Niu Z. An Anti-Freezing Hydrogel Electrolyte for Flexible Zinc-Ion Batteries Operating at $-70\text{ }^{\circ}\text{C}$. *Advanced Functional Materials* **33**, 2214546 (2023).
16. Varma AJ, Deshpande SV, Kennedy JF. Metal complexation by chitosan and its derivatives: a review. *Carbohydrate Polymers* **55**, 77-93 (2004).
17. Yan CY, Chen ZX, Huang H, Deng XY. Spontaneous Proton Chemistry Enables Ultralow-temperature and Long-life Aqueous Copper Metal Batteries. *Angewandte Chemie-International Edition* **62**, (2023).
18. Wang Y, *et al.* Competitive Coordination of Sodium Ions for High-Voltage Sodium Metal Batteries with Fast Reaction Speed. *Journal of the American Chemical Society* **146**, 7332-7340 (2024).
19. Liu Q, Yu Z, Zhuang Q, Kim J-K, Kang F, Zhang B. Anti-Fatigue Hydrogel Electrolyte for All-Flexible Zn-Ion Batteries. *Advanced Materials* **35**, 2300498 (2023).
20. Huang X, Zou Y, Jiang J. Electrochemical Oxidation of Glycerol to Dihydroxyacetone in Borate Buffer: Enhancing Activity and Selectivity by Borate–Polyol Coordination Chemistry. *ACS Sustainable Chemistry & Engineering* **9**, 14470-14479 (2021).
21. Khonina TyG, *et al.* Structural features and antimicrobial activity of hydrogels obtained by the sol–gel method from silicon, zinc, and boron glycerolates. *Journal of Sol–Gel Science and Technology* **95**, 682-692 (2020).
22. Bai C, *et al.* Influence of the pH in Reactions of Boric Acid/Borax with Simple Hydroxyl Compounds: Investigation by Raman Spectroscopy and DFT Calculations. *ChemistrySelect* **4**, 14132-14139 (2019).
23. Roget SA, Piskulich ZA, Thompson WH, Fayer MD. Identical Water Dynamics in Acrylamide Hydrogels, Polymers, and Monomers in Solution: Ultrafast IR Spectroscopy and Molecular Dynamics Simulations. *Journal of the American Chemical Society* **143**, 14855-14868 (2021).
24. Wang C, *et al.* Salt-tolerance training enabled flexible molten hydrate gel electrolytes for energy-dense and stable zinc storage. *Matter* **6**, 3993-4012 (2023).
25. Liu T, *et al.* A Bio-Inspired Methylation Approach to Salt-Concentrated Hydrogel Electrolytes for Long-Life Rechargeable Batteries. *Angewandte Chemie International Edition* **62**, e202311589 (2023).

REVIEWER COMMENTS

Reviewer #1 (Remarks to the Author):

The reviewer is satisfied with the data provided in the response letter. However, some data are not thoroughly discussed. One point that may require detailed discussion is the uncoordinated anion. The author demonstrated that the reconstructed structures of the anion exhibit lower solvation energy and a narrower energy gap compared to $\text{BF}_4\text{-(H}_2\text{O)}_a\text{(C}_3\text{H}_8\text{O}_3)_b\text{(H}_3\text{BO}_3)_c\text{(PAM)}_d\text{(CS)}_e$ ($abcde=30000$), indicating an accelerated charge transfer process. However, the differences between other complexes, apart from 30000, are not explained. For instance, 21000 shows the second-highest energy gap of 7.17 eV in Fig. R4, 10101 represents the lowest, and 20010 presents the highest solvation energy in Fig. R4. The upper and lower limits of the displayed energy levels in Fig. R5 for different complexes should be clearly specified.

Reviewer #2 (Remarks to the Author):

The authors L. Yao, J. Liu, F. Zhang, B. Wen, X. Chi, and Y. Liu provided comprehensive, detailed, and well-thought answers to all remarks from my previous revision. The article is significantly improved and ready for publication.

Reviewer #3 (Remarks to the Author):

Thank you for the comprehensive response and the effort to address all reviewer's comments. After carefully reading the revised manuscript, only a few minor remarks come to mind:

- Please add the computational level to the figure captions you show DFT results.
- As a suggestion (totally optional), consider moving Supplementary Fig. 10 to the main text.

Response to Reviewers' Comments

Reviewer #1 (Remarks to the Author):

The reviewer is satisfied with the data provided in the response letter. However, some data are not thoroughly discussed. One point that may require detailed discussion is the uncoordinated anion. The author demonstrated that the reconstructed structures of the anion exhibit lower solvation energy and a narrower energy gap compared to $\text{BF}_4^- (\text{H}_2\text{O})_a(\text{C}_3\text{H}_8\text{O}_3)_b(\text{H}_3\text{BO}_3)_c(\text{PAM})_d(\text{CS})_e$ ($abcde=30000$), indicating an accelerated charge transfer process. However, the differences between other complexes, apart from 30000, are not explained. For instance, 21000 shows the second-highest energy gap of 7.17 eV in Fig. R4, 10101 represents the lowest, and 20010 presents the highest solvation energy in Fig. R4. The upper and lower limits of the displayed energy levels in Fig. R5 for different complexes should be clearly specified.

Our response: We appreciate the reviewer for approving the importance and novelty of our work and constructive revision advice. We have addressed the reviewer's comments point by point.

1. The author demonstrated that the reconstructed structures of the anion exhibit lower solvation energy and a narrower energy gap compared to $\text{BF}_4^- (\text{H}_2\text{O})_a(\text{C}_3\text{H}_8\text{O}_3)_b(\text{H}_3\text{BO}_3)_c(\text{PAM})_d(\text{CS})_e$ ($abcde=30000$), indicating an accelerated charge transfer process. However, the differences between other complexes, apart from 30000, are not explained. For instance, 21000 shows the second-highest energy gap of 7.17 eV in Fig. R5, 10101 represents the lowest, and 20010 presents the highest solvation energy in Fig. R4.

Our response: Thanks for your constructive question. Firstly, for the solvation behavior of free anions tuned by the CRACSS strategies as shown in Fig. R4, the 10101 exhibited the highest solvation energy due to the more abundant hydrogen-bond donors (-NH₂, -OH) of CS than PAM with only -CONH₂ donors. In the presence of H₂O and

PAM, the single coordination sites make their interaction with BF_4^- relatively weak. Meanwhile, due to the presence of PAM, the coordination of H_2O will be limited, resulting in a less stable solvation structure than the strong coordination from glycerol and CS. Notably, all these structures are important for the interfacial stabilization of Zn anodes and the desolvation process for the PANI cathodes.

Fig. R4 | Solvation energy of different solvation structures of uncoordinated anions denoted as $\text{BF}_4^-(\text{H}_2\text{O})_a(\text{C}_3\text{H}_8\text{O}_3)_b(\text{H}_3\text{BO}_3)_c(\text{PAM})_d(\text{CS})_e$.

What's more, for the 21000 mentioned in Fig. R5, the electron states in the HOMO level are mainly occupied by the H_2O molecule. In contrast, the LUMO level is contributed from H_3BO_3 since the typical electron deficient characteristics in H_3BO_3 because of the sp^2 hybridization. On the other hand, the LUMO level of $\text{C}_3\text{H}_8\text{O}_3$ is mainly contributed by anti-bonding orbitals with relatively low energy levels of $-\text{OH}$ groups, whereas the HOMO level of $\text{C}_3\text{H}_8\text{O}_3$ is relatively higher due to the lone-pairs electrons in $-\text{OH}$ groups. Thus, the solvation structures such as 10110 and 01110 consisting of $\text{C}_3\text{H}_8\text{O}_3$ exhibit lower energy gaps. As for the PAM-involved structures, the relatively higher HOMO can be attributed to the non-bonded electronic state of N and O in $-\text{CONH}_2$ groups from PAM. Similarly, the HOMO of CS is relatively higher due to the sufficient $-\text{NH}_2$ and $-\text{OH}$ groups compared with PAM, which ensures the solvation structures involved with CS exhibit lower energy gaps.

In conclusion, the mechanism of reduction of energy gaps of all the solvation structures tuned by the CRACSS strategy can be clarified from the prospective of electronic structures and energy levels. The CRACSS strategy paves a new way in electrolyte design for AZMBs with fast kinetics and stable interface for wide temperature range applications. Thank you so much again for your constructive and

thoughtful questions.

Fig. R5 | Energy level of different solvation structures of uncoordinated anions denoted as $\text{BF}_4^-(\text{H}_2\text{O})_a(\text{C}_3\text{H}_8\text{O}_3)_b(\text{H}_3\text{BO}_3)_c(\text{PAM})_d(\text{CS})_e$.

In revision: The sentence ‘For the solvation behavior of free anions tuned by the CRACSS strategies, the 10101 exhibited the highest solvation energy due to the more abundant hydrogen-bond donors (-NH₂, -OH) of CS than PAM with only -CONH₂ donors. In the presence of H₂O and PAM, the single coordination sites make their interaction with BF₄⁻ relatively weak. Meanwhile, due to the presence of PAM, the coordination of H₂O will be limited, resulting in a less stable solvation structure than the strong coordination from glycerol and CS. Notably, the two structures are important for the interfacial stabilization of Zn anodes and desolvation process for the PANI cathodes.’ was added to the Supplementary Fig. 16 to further clarify the differences in solvation energies from different complexes.

The sentence ‘For the 21000 mentioned in Supplementary Fig. 39, the electron states in the HOMO level are mainly occupied by the H₂O molecule. In contrast, the LUMO level is contributed from H₃BO₃ since the typical electron deficient characteristics in H₃BO₃ because of the sp₂ hybridization. On the other hand, the LUMO level of C₃H₈O₃ is mainly contributed by anti-bonding orbitals with relatively low energy levels of -OH groups, whereas the HOMO level of C₃H₈O₃ is relatively higher due to the lone-pairs electrons in -OH groups. Thus, the solvation structures such as 10110 and 01110 consisting of C₃H₈O₃ exhibit lower energy gaps. As for the PAM-involved structures, the relatively higher HOMO can be attributed to the non-bonded electronic state of N and O in -CONH₂ groups from PAM. Similarly, the HOMO of CS

is relatively higher due to the more sufficient -NH₂ and -OH groups compared with PAM, which ensures the solvation structures involved with CS exhibit lower energy gaps. In conclusion, the mechanism of reduction of energy gaps of all the solvation structures tuned by the CRACSS strategy can be clarified from the prospective of electronic structures and energy levels.' was added to the Supplementary Fig. 39 to further clarify the differences in energy levels from different complexes.

Reviewer #2 (Remarks to the Author):

The authors L. Yao, J. Liu, F. Zhang, B. Wen, X. Chi, and Y. Liu provided comprehensive, detailed, and well-thought answers to all remarks from my previous revision.

The article is significantly improved and ready for publication.

Our response: We appreciate the reviewer for recognizing the novelty and importance of our work. Thanks to your insightful advice, we hope the CRACSS strategy could pave a new way and create inspiration for researchers aiming at electrolyte design for AZMBs for wide temperature range applications.

Reviewer #3 (Remarks to the Author):

Thank you for the comprehensive response and the effort to address all reviewer's comments. After carefully reading the revised manuscript, only a few minor remarks come to mind:

- Please add the computational level to the figure captions you show DFT results.*
- As a suggestion (totally optional), consider moving Supplementary Fig. 10 to the main text.*

Our response: We appreciate the reviewer for recognizing the innovations of our work and proposing very constructive revision advice. We have carefully and thoroughly addressed the reviewer's comments below.

- 1. Please add the computational level to the figure captions you show DFT results.*

Our response: Thanks for your sincere advice. The mark '(B97-3c)' of all the calculations about solvation structures using B97-3c is added to the captions of figures.

- 2. As a suggestion (totally optional), consider moving Supplementary Fig. 10 to the main text.*

Our response: Thanks for your sincere advice. It's meaningful to move Supplementary Fig. 10 to the main text, but it's difficult to make a suitable replacement due to the length limit of the article page. Thus, we decided to keep Supplementary Fig. 10 where it used to be.